# EDiT: A Local-SGD-Based Efficient Distributed Training Method for Large Language Models

**Jialiang Cheng, Ning Gao, Yun Yue, Zhiling Ye, Jiadi Jiang, Jian Sha** *
Ant Group
{jichen.cjl,yunsheng.gn,yueyun.yy}@antgroup.com
{yezhiling.yzl,jiadi.jjd,jian.sha}@antgroup.com

## ABSTRACT

Distributed training methods are crucial for large language models (LLMs). However, existing distributed training methods often suffer from communication bottlenecks, stragglers, and limited elasticity, particularly in heterogeneous or large-scale environments. Local SGD methods have been proposed to address these issues, but their effectiveness remains limited to small-scale training due to additional memory overhead and lack of concerns on efficiency and stability. To tackle these issues, we propose **EDiT**, an innovative **E**fficient **Di**stributed **T**raining method that combines a tailored Local SGD approach with model sharding techniques to enhance large-scale training efficiency. EDiT performs layer-wise parameter synchronization during forward pass, reducing communication and memory overhead and enabling overlap. Besides, EDiT employs a pseudo gradient penalty strategy to suppress loss spikes, which ensures training stability and improves performance. Additionally, we introduce **A-EDiT**, a fully asynchronous variant of EDiT that accommodates heterogeneous clusters. Building on EDiT/A-EDiT, we conduct a series of experiments to validate large-scale asynchronous training for LLMs, accompanied by comprehensive analyses. Experimental results demonstrate the superior performance of EDiT/A-EDiT, establishing them as robust solutions for distributed LLM training in diverse computational ecosystems. The code is available at Atorch codebase [1].

## 1 INTRODUCTION

With the explosive growth of model scale and data volume (Touvron et al., 2023; Bai et al., 2023), distributed methods (Rajbhandari et al., 2020; Narayanan et al., 2021; Dean et al., 2012) become increasingly critical for training deep neural networks. These approaches rely on synchronous paradigm, which introduces significant communication overhead during the training process (Douillard et al., 2023). Besides, the synchronous paradigm also introduces the straggler problem, where faster workers are idle waiting for the slower ones to catch up. This issue is particularly prevalent in large/heterogeneous clusters (Lian et al., 2018). Lastly, in resource-constrained clusters, there is a compelling need for elastic training (Li et al., 2023). However, synchronous training paradigms struggle in elastic settings, where dynamic scaling of resources disrupts optimal hyperparameters.

These challenges have spurred significant research into distributed optimization methods. A typical method is Local Stochastic Gradient Descent (a.k.a Local SGD or Local-Update SGD) (Zhang et al., 2016), where each worker independently executes multiple local optimization steps in parallel before averaging model parameters across all workers. Subsequent studies have improved upon this foundational paradigm to improve the performance (Lin et al., 2019; Wang et al., 2019; Douillard et al., 2023). However, existing Local SGD methods are not easily applicable to the training of large language models (LLMs). These methods do not handle model sharding well, preventing their application to models larger than billions of parameters. Moreover, they have focused on small-

---

*Corresponding author.
[1] https://github.com/intelligent-machine-learning/atorch/

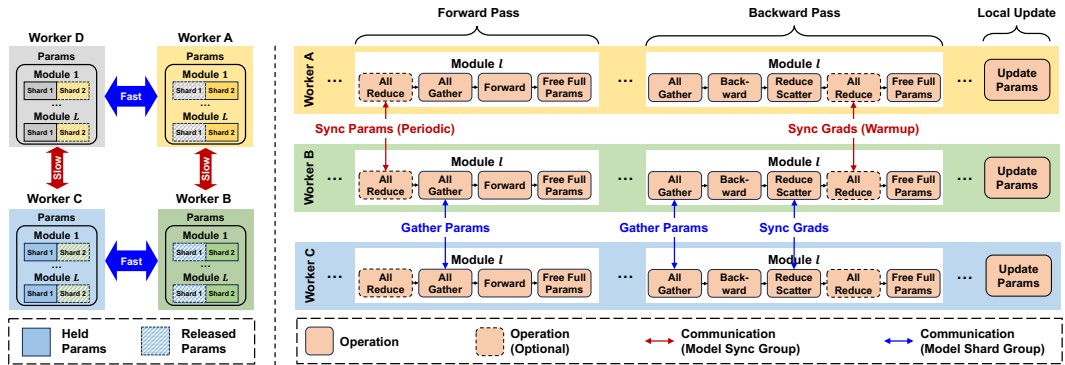

Figure 1: The schematic illustration of our proposed EDiT method with $4$ workers and a $2 \times 2$ device mesh as an example. The left part shows the communication groups and parameter sharding, and the right part presents the detailed computation and communication flows within worker B.

scale, highly curated datasets (Zhang et al., 2016; Douillard et al., 2023), making their results less transferable to LLM training that relies on vast, noisy datasets where instability may be introduced during training. Besides, although current Local SGD methods diminish the impact of random stragglers, they still struggle with the presence of consistent stragglers within heterogeneous devices (Liu et al., 2024). Additionally, in most existing Local SGD methods, parameter synchronization operations will introduce non-overlapped communication overhead (Sun et al., 2023). Lastly, current Local SGD methods predominantly employ a uniform averaging strategy to synchronize the parameters, failing to fully capitalize on the inherent differences in training progress across diverse workers (Douillard et al., 2023).

To address these challenges, we propose a novel Efficient Distributed Training (EDiT) method for large language models. As illustrated in Figure 1, EDiT employs a hierarchical distribution strategy on a two-dimensional device mesh, where all workers are data parallel. Model parameters are fully sharded along the model shard dimension and synchronized along the model sync dimension. With the efficient communication links within the model shard groups and the low-frequency periodic synchronization strategy within the model sync groups, the impact of communication overhead and random stragglers is effectively alleviated. When synchronizing parameters, EDiT operates layer by layer during the forward pass and makes use of a prefetch strategy to overlap computation and communication, thereby reducing the additional communication and GPU memory overhead introduced by parameter synchronization. Additionally, EDiT employs a novel pseudo-gradient penalty method, which addresses the instability problem caused by diverse large-scale corpus and leverages the differences among workers to improve performance. Furthermore, we propose an asynchronous variant of the EDiT method named A-EDiT to deal with the consistent stragglers in heterogeneous clusters. We conducted a comprehensive evaluation of our proposed methods on LLM tasks, demonstrating its effectiveness compared to state-of-the-art methods.

Our primary contributions can be summarized as follows:

- Engineering Innovation: We introduce EDiT, an efficient large-scale distributed training method that integrates Local SGD with the model sharding strategy. EDiT reduces the impact of stragglers and communication overhead and supports elastic training.

- Algorithmic Novelty: EDiT performs layer-wise parameter sync during forward pass to reduce communication and memory overhead. With prefetch strategy, the parameter-sync communication can be further overlapped with computation. Besides, we propose a new pseudo gradient penalty method to improve the training stability and model performance. We also provide a fully asynchronous variant of EDiT, called A-EDiT, to address the challenges of consistent stragglers.

- Practical Contributions: We provide a large-scale verification of asynchronous pre-training for LLMs, along with an extensive analysis of convergence, generalization, acceleration, scalability, and stability. This work offers critical insights into optimizing asynchronous distributed LLM training at scale.

## 2 RELATED WORK

One of the early works that proposed the concept of Local SGD was Zhang et al. (2016), establishing the paradigm of parallel multi-step training followed by periodic averaging. Lin et al. (2019) introduced the Post Local SGD method, which starts with standard synchronized training for warm-up before switching to the Local SGD mode. SlowMo (Wang et al., 2019) utilizes a slow momentum to transform model averaging into moving average. DiLoCo (Douillard et al., 2023) demonstrates that the Nesterov optimizer (Nesterov, 1983) is suitable as an outer optimizer. Multi-Level Local SGD (Castiglia et al., 2020) partition the network into disjoint sub-networks and hierarchically synchronizes the models. Wang & Joshi (2019) and Balles et al. (2023) have respectively explored the optimal hyperparameter settings for Local SGD. Shen et al. (2021) advocated for gradually increasing synchronization intervals while decreasing learning rates to optimize model performance. Extensive theoretical analyses of Local SGD have also emerged. Yu et al. (2019), Khaled et al. (2020), Spiridonoff et al. (2020), and Deng et al. (2022) examined convergence rates under various conditions. Gu et al. (2022) found that Local SGD improves generalization with a small learning rate and long training duration. Pan & Song (2023) demonstrated faster convergence by leveraging second-order information.

Researchers have also explored the combination of Local SGD with asynchronous training paradigms that decouple computation and communication. Early works were predominantly based on the federated learning framework (Xie et al., 2019). FedBuff (Nguyen et al., 2022) updates the server model only after accumulating a certain amount of pseudo gradients. DN-DyLN (Liu et al., 2024) improves the buffer mechanism to employ delayed Nesterov update. TimelyFL (Zhang et al., 2023) dynamically adjusts the local training workload according to the real-time resource situation. Subsequently, several works based on other architectures were also proposed. Gossip-PGA (Chen et al., 2021) incorporates periodic global averaging into the gossip SGD framework (Lian et al., 2017). CO2 (Sun et al., 2023) utilizes Local SGD and asynchronous communication to hide the overhead. A key challenge for asynchronous training is the staled model problem, resulting in inferior performance compared to synchronous training methods (Liu et al., 2024).

Notably, current All-Reduce-based Local SGD methods (Lin et al., 2019; Wang et al., 2019; Sun et al., 2023) hold complete model parameters on each GPU, making it difficult to handle model sharding for LLM training. Although Sun et al. (2023) claims that they can combine CO2 with ZeRO series optimizers (Rajbhandari et al., 2020), the additional communication introduced degrades CO2 to a synchronized mode, negating the performance gains from periodic synchronization and overlapped communication. Furthermore, the extra parameters and outer momentum further increase memory pressure, limiting their scalability to larger models. In contrast, our proposed EDiT and A-EDiT methods effectively utilize the characteristics of model sharding, leveraging device mesh, layer-wise parameter synchronization, prefetch strategy, and CPU offload to minimize communication and memory overhead, making it more suitable for LLM training.

## 3 METHOD

### 3.1 OVERVIEW

Our proposed EDiT method integrates model sharding with periodic synchronization to accelerate the training of LLMs. The detailed procedure of EDiT is illustrated in Figure 1 and formally outlined in Algorithm 1 in Appendix. To start with, EDiT builds an $M \times N$ device mesh across $K$ workers : $M$ model sync groups $\mathcal{G}^r = \{\mathcal{G}_1^r, \cdots, \mathcal{G}_M^r\}$ with each comprising $N$ workers $\mathcal{G}_i^r = \{\mathcal{W}_{(i,1)}, \mathcal{W}_{(i,2)}, \cdots, \mathcal{W}_{(i,N)}\}$ [2], and $N$ model shard groups $\mathcal{G}^s = \{\mathcal{G}_1^s, \cdots, \mathcal{G}_N^s\}$ with each comprising $M$ workers $\mathcal{G}_i^s = \{\mathcal{W}_{(1,i)}, \mathcal{W}_{(2,i)}, \cdots, \mathcal{W}_{(M,i)}\}$, where $M \times N = K$. This structured arrangement aims to tailor communication patterns to the diverse capabilities and network latencies inherent in the distributed system. For instance, in a multi-node GPU cluster where intra-node communication is significantly faster than inter-node communication, all GPUs within the same node can be connected as a model shard group, while GPUs of the same rank across different nodes can be connected as a model sync group. Model parameters are sharded uniformly in each model

---

[2]The employment of double subscripts herein is merely a notational convenience to denote the relationship between workers and groups. Similar considerations apply to the cases discussed below.

shard group and each worker $\mathcal{W}_i$ retains a fraction of each parameter for the complete $L$ modules: $\boldsymbol{\theta}^{(i)} = \{\boldsymbol{\theta}^{(i,1)}, \cdots, \boldsymbol{\theta}^{(i,L)}\}$. In this way, workers as a whole within a model shard group $\mathcal{G}_i^s$ maintain a complete replica of model parameters: $\boldsymbol{\theta} = \text{Concat}(\{\boldsymbol{\theta}^{(i)} : \boldsymbol{\theta}^{(i)} \in \mathcal{G}_i^s\})$, while workers within a model sync group $\mathcal{G}_i^r$ maintain an identical shard of the parameters. The EDiT method centralizes communication-intensive operations within the model shard groups and utilizes periodic synchronization to mitigate the communication overhead in the model sync groups, thereby achieving training acceleration. To enhance the stability of the initial training process, EDiT utilizes a two-phase training strategy. This begins with a warmup phase using standard mini-batch SGD, followed by a periodic synchronization phase utilizing Local SGD. More specifically,

1. During the forward pass of the $l$-th module, if the current updated step requires model synchronization, *i.e.*, $(t*\tau+p) > t_{warm}$ and $p == 0$ where $t$ is the outer step, $p$ is the inner step, $t_{warm}$ is the number of warmup steps and $\tau$ is the synchronization interval, parameters are synced in model sync groups, as outlined in lines 7 to 9 in Algorithm 1. In practice, the communication overhead is minimal due to the large synchronization interval ($\tau \gg 1$) and sharded parameters. Herein a novel pseudo gradient penalty strategy is introduced to enhance training stability that will be detailed in Section 3.2. After that, each worker gathers the full module parameters through its model shard group for forward computations and promptly frees excess parameters to conserve memory.

2. During the backward pass of the $l$-th module, workers again aggregate parameters via model shard groups for gradient calculations, followed by a *reduce-scatter* operation to average gradients across each model shard group. If the current step $t$ is within the warmup phase, *i.e.*, $t \leq t_{warm}$, an additional *all-reduce* operation will be performed within each model sync group to synchronize gradients across all workers; otherwise this operation will be skipped (lines 19 to 21). Afterwards, each worker frees the excess parameters.

3. Once all modules have completed one forward-backward iteration, the optimizer updates the local parameters of each worker. Note that to distinguish from the outer optimizer (OuterOpt) used in parameter synchronization, we refer to the optimizer for local updates as the inner optimizer (InnerOpt).

Different from other Local SGD methodologies that synchronize parameters before next step, EDiT performs layer-wise parameter synchronization during forward pass. In practice, we normally employ a prefetch strategy that aggregates parameters for the upcoming module concurrently with ongoing computations, with which communications within model sync groups can be effectively overlapped with forward computations. In this way, EDiT further diminishes the additional communication overhead introduced by parameter synchronization.

It is also noteworthy that EDiT is compatible with most current large-scale distributed training frameworks. Although this manuscript mainly discusses its integration within ZeRO-3/FSDP framework (Rajbhandari et al., 2020), it can be transposed with relative ease to other frameworks such as 3D parallelism (Shoeybi et al., 2019).

## 3.2 PSEUDO GRADIENT PENALTY

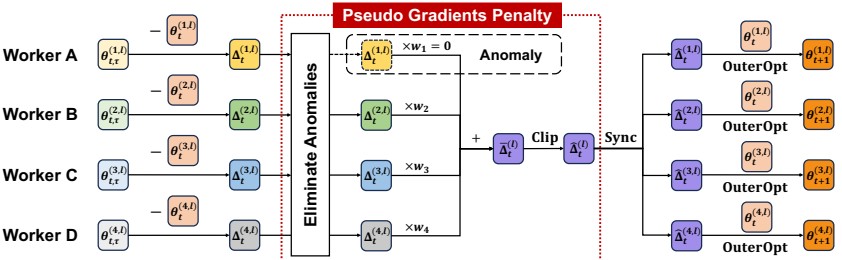

Figure 2: Illustration of model synchronization and our proposed pseudo gradient penalty method, depicted with an example of four workers in a model sync group.

Despite diligent data cleaning efforts, there are still significant amount of low-quality data in the LLM pre-training corpora (Albalak et al., 2024), resulting in training instability manifested as loss

spikes. This issue can be addressed by large batch sizes typical of synchronous training regimes, but becomes salient in Local SGD regimes where each worker operates on relatively smaller batches.

To tackle this issue, we introduce a novel pseudo gradient penalty strategy at the parameter synchronization stage, as depicted in Figure 2 and Algorithm 2 in Appendix. This strategy consists of anomaly elimination, weighted averaging, and gradient clipping. To illustrate the idea, we use a model sync group $\mathcal{G}_m^r = \{\mathcal{W}_1, \cdots, \mathcal{W}_N\}$ as an example. We begin by computing the pseudo gradients $\boldsymbol{\Delta}_t^{(i,l)} = \boldsymbol{\theta}_{t,\tau}^{(i,l)} - \boldsymbol{\theta}_t^{(i,l)}$ for each worker, where $\boldsymbol{\theta}_{t,\tau}^{(i,l)}$ is the sharded parameters of module $l$ held by worker $\mathcal{W}_i$ at outer step $t$ and inner step $\tau$, and $\boldsymbol{\theta}_t^{(i,l)}$ denotes the corresponding synchronized parameters at the beginning of outer step $t$.

**Anomaly elimination.** We first eliminate the significantly anomalous workers to reduce their adverse impacts on the overall model performance. Since anomalies cause substantial parameter fluctuations and lead to large pseudo-gradient norms, we use the pseudo-gradient norm as the criterion. Here we utilize an Exponential Moving Average (EMA) z-test method for statistical analysis. Let $G_t^{(i,l)} = \|\boldsymbol{\Delta}_t^{(i,l)}\|_2$ denotes the pseudo gradient norm for the worker $\mathcal{W}_i$, then the EMA z-score can be calculated by $z_t^{(i,l)} = \frac{G_t^{(i,l)} - \mu_t^{(i,l)}}{\sigma_t^{(i,l)}}$, where $\mu_t^{(i,l)}$ and $\sigma_t^{(i,l)}$ are the EMA mean and standard deviation of $G_t^{(i,l)}$, respectively. A worker $\mathcal{W}_i$ with $z_t^{(i,l)} > \delta$ is identified as an anomaly and its $G_t^{(i,l)}$ will be set to infinity, where $\delta$ is a threshold, typically set to 3 in practice. Both $\mu_t^{(i,l)}$ and $\sigma_t^{(i,l)}$ are updated at each step using an exponential moving average to capture the convergence trend of the gradient norm during the training process:

$$\mu_{t+1}^{(i,l)} = \alpha G_t^{(i,l)} + (1-\alpha)\mu_t^{(i,l)}, \quad \sigma_{t+1}^{(i,l)} = \sqrt{(1-\alpha)(\sigma_t^{(i,l)})^2 + \alpha(G_t^{(i,l)} - \mu_{t+1}^{(i,l)})^2}, \quad (1)$$

where $\alpha$ is a weighting coefficient, commonly assigned a value of $0.02$ in practical applications. The update of Equation 1 will be skipped if $G_t^{(i,l)}$ is infinite. In the preliminary stage, a warm-up period is set to establish stable values for $\mu_t^{(i,l)}$ and $\sigma_t^{(i,l)}$, during which no workers are flagged as anomalies. Notably, to maintain consistent updates within the same module, we compute the pseudo gradient norm for the entire module, and subsequently introduced gradient norm-related operations follow the same procedure. Because this process only introduces one scalar communication in the model shard groups, the overhead is negligible. If all workers are identified anomalous, all the parameters will be effectively rollbacked to the last synchronized parameters $\boldsymbol{\theta}_t^{(i,l)}$.

**Weighted averaging.** Furthermore, considering that large pseudo gradients may still exert considerable impacts on the overall update direction, we propose to weigh the pseudo gradients of each worker based on the norms, which was similarly demonstrated in Thakkar et al. (2023). The weight assigned to the pseudo gradients corresponding to $\mathcal{W}_i \in \mathcal{G}_m^r$ is calculated by

$$w_{t,i} = \frac{\exp(-G_t^{(i,l)})}{\sum_j \exp(-G_t^{(j,l)})}. \quad (2)$$

In this way, a larger pseudo gradient norm leads to stronger suppression, thereby allowing all workers to contribute equally to the update direction and thus increasing the likelihood to find the correct direction. Following that, by performing a weighted summation of all pseudo gradients in $\mathcal{G}_m^r$, we obtain the synchronized pseudo gradients:

$$\bar{\boldsymbol{\Delta}}_t^{(i,l)} = \sum_j w_{t,j}\boldsymbol{\Delta}_t^{(j,l)}, \forall \mathcal{W}_i \in \mathcal{G}_m^r. \quad (3)$$

**Gradient clip.** We then adopt a gradient clip strategy to constrain the update step size. Let $\bar{G}_t^{(i,l)} = \|\bar{\boldsymbol{\Delta}}_t^{(i,l)}\|_2$ denote the synchronized pseudo gradient norm and $\phi$ denote the threshold, the clip coefficient is computed by

$$\beta_t = \min(\phi/(\bar{G}_t^{(i,l)} + \epsilon), 1), \quad (4)$$

where $\epsilon$ is a small positive constant to avoid division by zero. The pseudo gradients are clipped by

$$\widehat{\boldsymbol{\Delta}}_t^{(i,l)} = \beta_t \bar{\boldsymbol{\Delta}}_t^{(i,l)}. \quad (5)$$

Proceeding further, we update $\boldsymbol{\theta}_t^{(i,l)}$ by $\boldsymbol{\theta}_{t+1}^{(i,l)} = \text{OuterOpt}(\boldsymbol{\theta}_t^{(i,l)}, \widehat{\boldsymbol{\Delta}}_t^{(i,l)})$ on each worker.

In EDiT, the extra parameters and outer momentum on each worker are sharded in correspondence with the sharded parameters. Compared to previous methods that maintain full extra parameters and outer momentum on each worker, EDiT reduces the additional memory usage. Additionally, based on layer-wise synchronization and prefetch strategy, EDiT can further offload the extra parameters and outer momentum to the CPU and only transfer the corresponding layer's data to GPU as needed, thereby further minimizing memory overhead. Since the data for each layer is relatively small, the GPU-CPU data transfer can be effectively overlapped with GPU computations and GPU-GPU communications, ensuring fast parameter synchronization.

## 3.3 ASYNCHRONOUS EDiT

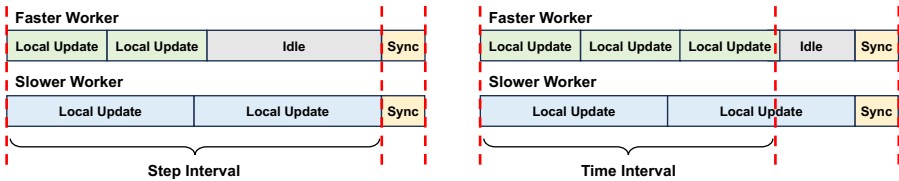

(a) The synchronization scheme of EDiT.    (b) The synchronization scheme of A-EDiT.

Figure 3: A comparison of the synchronization scheme of EDiT and A-EDiT.

EDiT requires periodic synchronization at every $\tau$ inner iterations. However, the fastest worker idles awaiting the peers to finish $\tau$ iterations even if it completes its own $\tau$ iterations earlier. As a consequence, the overall training efficiency is pegged to the slowest worker. This issue becomes more pronounced in heterogeneous clusters, where nodes are equipped with diverse devices.

Intuitively, it would be beneficial to allow different workers to train at their own pace and remove the constraint of fixed-step synchronization. Therefore, we propose an asynchronous variant of the EDiT method, named A-EDiT. The differences are depicted in Figure 3. Herein, we set a fixed time interval $\tau_{time}$, and let each worker update locally until surpassing this specified time threshold. Then, a parameter synchronization ensues. This modification enables faster workers to undertake more iterations in each inner loop. Theoretically, no worker will wait longer than the single step time of the slowest worker at each parameter synchronization. We empirically verified that A-EDiT achieves faster training in all scenarios with comparable model performance.

## 4 EXPERIMENTS

### 4.1 EXPERIMENTAL SETUPS

**Models** We consider four different scales of Llama models (Touvron et al., 2023) in our experiments: 350M, 1B, 3B, and 7B. Their specific configurations are detailed in Table 3 in Appendix.

**Datasets** We use a new large-scale open-source dataset, FineWeb-Edu (Lozhkov et al., 2024) in our experiments. Additionally, we also utilize an in-house private out of production pre-training dataset, which we will refer to as in-house dataset below.

**Baselines** We consider several state-of-the-art methods, including standard mini-batch (Baseline), Post Local SGD (Lin et al., 2019), DiLoCo (Douillard et al., 2023), and CO2/CO2* (Sun et al., 2023). Here CO2* is the memory-efficient version of CO2 that shards extra parameters and outer momentum across workers (Sun et al., 2023).

**Training** Following DiLoCo (Douillard et al., 2023), we use AdamW (Loshchilov & Hutter, 2019) as the inner optimizer and Nesterov momentum (Nesterov, 1983) as the outer optimizer. The models are initialized with $\mu$P (Yang et al., 2021) for efficient hyperparameter search. Synchronization intervals $\tau$ and $\tau_{time}$ are set to 128 and 600s, respectively. Experiments are conducted on eight Nvidia A100 GPU nodes with 64 GPUs and an $8 \times 8$ device mesh. $\phi$ is 10 for pseudo gradient clip.

For more detailed setups, please refer to the Appendix A.2.

## 4.2 CONVERGENCE AND GENERALIZATION

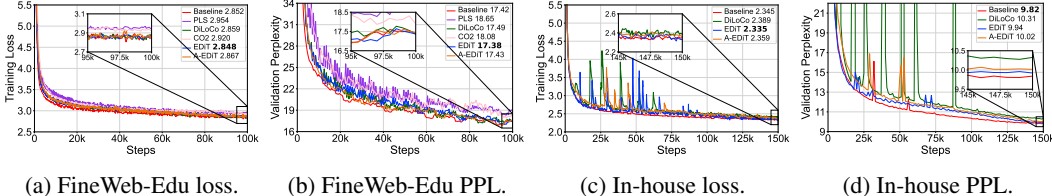

| (a) FineWeb-Edu loss. | (b) FineWeb-Edu PPL. | (c) In-house loss. | (d) In-house PPL. |

Figure 4: The loss and PPL curves of different methods on the (a) & (b) FineWeb-Edu dataset and (c) & (d) in-house dataset. The final values are marked, with the best ones in bold. Here we use the average of the last 10 values as results to prevent randomness. PLS is short for Post Local SGD.

Table 1: The evaluation results for different methods on the public benchmarks (Fourrier et al., 2023; OpenCompass Contributors, 2023), with the best results in bold and second-best underlined. PLS is short for Post Local SGD.

| Benchmark | FineWeb-Edu dataset | | | | | | in-house dataset | | | |
| | Baseline | PLS | DiLoCo | CO2 | EDiT | A-EDiT | Baseline | DiLoCo | EDiT | A-EDiT |
|---|---|---|---|---|---|---|---|---|---|---|
| MMLU (↑) | 32.28 | 30.86 | **32.55** | 31.33 | 32.29 | 31.96 | 24.12 | **24.63** | 24.47 | 24.56 |
| ARC-E (↑) | 59.90 | 57.60 | 58.60 | 57.00 | **60.00** | 57.70 | 36.80 | 36.70 | **38.70** | 37.40 |
| ARC-C (↑) | 30.20 | 28.60 | 31.00 | 30.50 | **32.40** | 30.20 | 22.50 | 22.80 | **23.00** | 22.40 |
| HellaSwag (↑) | 50.99 | 48.03 | 51.64 | 48.66 | **51.75** | 51.60 | 40.60 | 40.80 | **40.90** | 40.20 |
| PIQA (↑) | **69.90** | 67.80 | 69.50 | 67.00 | 68.10 | **69.90** | **67.10** | 66.80 | 67.00 | 66.40 |
| CommonSense-QA (↑) | **37.40** | 33.80 | 35.40 | 34.40 | 36.30 | 35.30 | **18.50** | 18.20 | **18.50** | 17.90 |
| OpenBookQA (↑) | 25.40 | 22.80 | 25.20 | 24.40 | **26.00** | 24.00 | 18.00 | 17.80 | 18.00 | **18.20** |
| WinoGrande (↑) | 50.70 | 49.20 | 47.80 | 50.70 | **51.70** | 50.50 | 49.10 | **49.20** | 49.10 | 48.80 |
| Average (↑) | 44.60 | 42.34 | 43.96 | 43.00 | **44.82** | 43.90 | 34.59 | 34.62 | **34.96** | 34.49 |

We first applied different methods to train the Llama 1B model on the FineWeb-Edu dataset and in-house dataset separately. Here we only compared the best-performing methods, *i.e.*, Baseline, DiLoCo, EDiT, and A-EDiT, on the in-house dataset. The training loss (↓) [3] and validation PPL (↓) results are shown in Figure 4.

As can be seen, our proposed EDiT and A-EDiT both achieve consistently good performance. Specifically, EDiT achieves the lowest training loss on both datasets and achieves the lowest validation PPL on the FineWeb-Edu dataset, even surpassing the Baseline. A-EDiT marginally lags behind the sync version due to the lagging workers, but it still performs better than other methods in most scenarios. Because the in-house dataset contains diverse data types and lower-quality corpora, DiLoCo (Douillard et al., 2023) experienced a noticeable decline in performance. In contrast, EDiT and A-EDiT filtered out low-quality data with the pseudo gradient penalty strategy, achieving results that were nearly comparable to the Baseline.

We evaluated the trained models on public benchmarks (Fourrier et al., 2023; OpenCompass Contributors, 2023). Table 1 presents the evaluation results. As can be seen, the models trained with EDiT both achieve the best average performance, and A-EDiT also performs well on the eight evaluation benchmarks. These results demonstrate that both EDiT and A-EDiT exhibit strong convergence and generalization capabilities.

Besides, we additionally trained the Llama 350M, 3B, and 7B models on the FineWeb-Edu dataset using EDiT, the results in Figure 8 and Table 5 demonstrate that EDiT performs consistently well across different model scales.

## 4.3 ACCELERATION

We measured the speeds of different methods when training Llama models of four different scales on two A100 nodes. The synchronization interval was set to 5, and the results are the average

---

[3]In this manuscript the ↑ means the bigger the better and the ↓ means the smaller the better.

Table 2: The speeds of different methods on training models of various scales. The values in the table correspond to throughput (tokens/sec) and TFLOPS, respectively.

|  | Baseline | Post Local SGD | DiLoCo | CO2 | CO2* | EDiT | A-EDiT |
|---|---|---|---|---|---|---|---|
| 350M | 4.52e5/107 | 4.67e5/111 | 4.56e5/108 | 4.84e5/116 | 4.66e5/110 | 4.81e5/114 | 4.82e5/115 |
| 1B | 2.08e5/146 | 2.12e5/149 | 1.87e5/131* | OOM | 2.12e5/148 | 2.25e5/158 | 2.27e5/160 |
| 3B | 1.05e5/177 | OOM | OOM | OOM | OOM | 1.11e5/187 | 1.12e5/189 |
| 7B | 5.14e4/200 | OOM | OOM | OOM | OOM | 5.42e4/211 | 5.45e4/213 |

throughput (tokens/sec) and TFLOPS over 100 steps. As shown in Table 2, all Local SGD-based methods achieved higher throughput than the Baseline. It should be noted that when training the Llama 1B model with DiLoCo, extra parameters and outer momentum were placed on CPUs to prevent out of memory (OOM), resulting in non-overlapped extra GPU-CPU data transfer overhead. While CO2 achieved the highest throughput on the smallest model, holding its extra parameters and outer momentum caused significant memory overhead preventing the method to be scaled beyond 350M model. CO2* alleviates memory pressure by sharding extra parameters and outer momentum, but introduces additional non-overlapping communication, causing a throughput drop. Our proposed methods synchronize sharded parameters layer-by-layer during the forward pass, and utilize a prefetch strategy to overlap computation with communication, achieving nearly the same throughput as CO2 ($-0.5\%$). We also performed a profiling analysis of the synchronization operations for different methods, and detailed results can be found in Appendix A.3.2 and Figure 9.

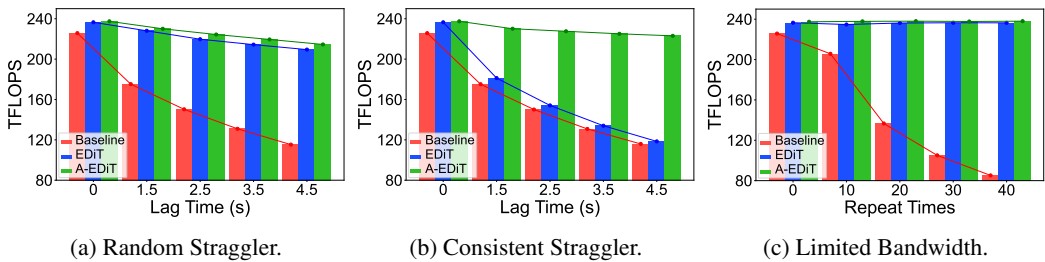

(a) Random Straggler.      (b) Consistent Straggler.      (c) Limited Bandwidth.

Figure 5: The TFLOPS of different methods under different training scenarios.

We further evaluated the training speed of our proposed EDiT and A-EDiT methods against the Baseline method in various more challenging training scenarios. Here we manually introduced stragglers and communication delays. Specifically, we simulated stragglers by pausing the training process of one selected node at each step, and simulated inter-node bandwidth constraints by artificially repeating inter-node communications. Experiments were conducted on the Llama 7B model. Detailed experimental results are presented in Figure 5 and Table 6 in Appendix.

The results reveal a consistent trend where A-EDiT and EDiT outperforms the Baseline method. As anticipated, the Baseline's training speed declines rapidly with increased lag time or inter-node congestion. In the random straggler scenario, EDiT and A-EDiT experience only slight speed reductions. This is attributed to the periodic synchronization that ensures relatively uniform training speeds across workers. In the consistent straggler scenario, since the cumulative delay at a single node cannot be eliminated by periodic synchronization, the performance of EDiT declines visibly. A-EDiT, leveraging its asynchronous nature, maintains a high training speed. In the bandwidth-constrained scenario, both EDiT and A-EDiT are not affected. This is due to the large synchronization interval, which minimizes inter-node communication overhead. In summary, our proposed methods consistently demonstrate superior training speed compared to the Baseline method across various scenarios, and A-EDiT further effectively addresses the issue of consistent stragglers.

## 4.4 SCALABILITY

Elastic training is the ability to dynamically adjust the resources in accordance with workload fluctuations. However, varying the resources alters the global batch size and requires additional learning rate tuning. Intuitively, the optimal learning rate of the Local SGD methods may be solely related to

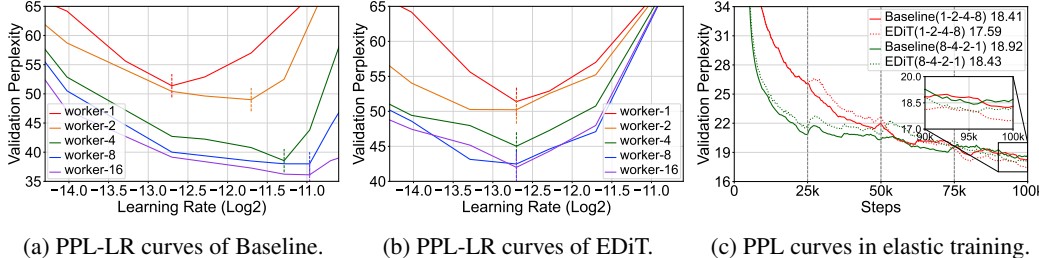

(a) PPL-LR curves of Baseline.  (b) PPL-LR curves of EDiT.  (c) PPL curves in elastic training.

Figure 6: (a) & (b) The PPL results against learning rates ($\log 2$ scale) under different number of workers for the Baseline and EDiT methods. (c) The PPL curves in the simulated training scenarios.

the per-worker batch size, which has not been extensively studied in prior research. To validate this hypothesis, we conducted experiments on the Llama 350M model to investigate the optimal learning rate shift for the Baseline method and EDiT method under different worker numbers, fixing the batch size per worker at 128. The validation PPL results are shown in Figure 6a and Figure 6b, and the detailed training losses are shown in Figure 10 in Appendix. It can be seen that as the worker number increases, the optimal learning rate for Baseline gradually increases, whereas that for EDiT consistently remains at 1.5e-4. These results validate our hypothesis. The scalability of EDiT makes it suitable for elastic training scenarios. Besides, this property enables us to economize resources by initially tuning the learning rate on a single worker before scaling up to hundreds of workers. We also note that the training loss for EDiT is more stable than that of the Baseline method across different worker numbers and learning rates, as shown in Figure 10. This not only demonstrates the robustness of EDiT but also highlights its potential to maintain consistent performance across diverse training configurations.

We further simulated a realistic elastic training scenario. We conducted experiments on the Llama 1B model, setting the batch size per worker to 128 and fixing the learning rate at 1.5e-4. We systematically scaled the worker number upwards (1-2-4-8) and downwards (8-4-2-1), training for 25,000 steps at each worker number, and observed the validation PPL for both the Baseline and EDiT methods. As illustrated in Figure 6c, although the Baseline method initially decreases faster than EDiT, EDiT maintains a significant decline rate in the later stages and achieves the optimal PPL values in both scaling scenarios, yielding a $4.5\%$ and $2.6\%$ improvement, respectively. These findings affirm the EDiT's viability and advantage in real-world, elastic training scenarios.

## 4.5 ABLATION STUDY

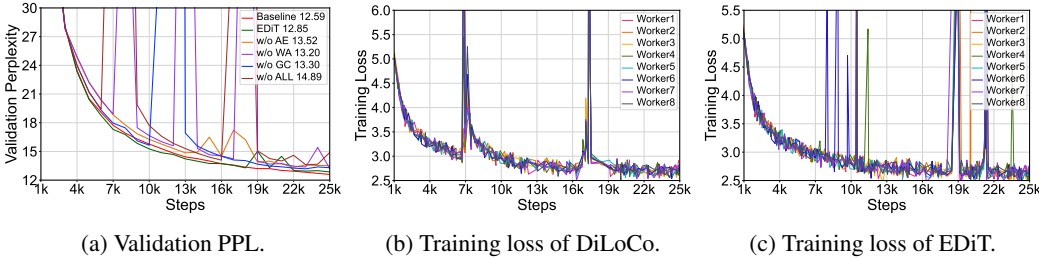

(a) Validation PPL.  (b) Training loss of DiLoCo.  (c) Training loss of EDiT.

Figure 7: (a) The validation PPL curves of different versions of EDiT with the final PPL values marked. (b) & (c) The training loss curves for DiLoCo and EDiT, respectively.

We conducted ablation studies on the pseudo gradient penalty strategy to better understand its capabilities. In this experiment, we employ the in-house dataset as it is of higher diversity and thus serves as an ideal testbed. We individually removed anomaly elimination (w/o AE), weighted averaging (w/o WA), and gradient clip (w/o GC) from EDiT, as well as all three components simultaneously (w/o ALL). The validation PPL results are shown in Figure 7a. It can be observed that without the pseudo gradient penalty strategy (w/o ALL), the PPL curve exhibits noticeable spikes and deviates considerably from the Baseline. Individually removing anomaly elimination, weighted averaging,

or gradient clip each adversely affects stability and validation PPL, demonstrating that every component of the pseudo gradient penalty strategy is effective. We further investigated the training losses across eight different workers. As depicted in Figure 7b and Figure 7c, all workers in DiLoCo simultaneously encounter loss spikes and take a long time to recover. In contrast, EDiT can swiftly rectify deviations in individual workers. Even when all workers experience abnormal losses, they can promptly revert to normal loss levels through the rollback strategy. These results demonstrate the effectiveness of the pseudo gradient penalty strategy.

## 5 THEORETICAL ANALYSIS

In this section, we choose SGD (Robbins & Monro, 1951) as the inner optimizer and the outer optimizer for simplicity. Under the framework developed in Wang et al. (2019), we have the following convergence theorem.

**Theorem 1.** *Suppose that the following assumptions are satisfied:*

1. *$\mathcal{L}$ is differential and lower bounded, i.e., $\mathcal{L}(\boldsymbol{\theta}^*) > -\infty$ where $\boldsymbol{\theta}^*$ is an optimal solution. $\mathcal{L}$ is also L-smooth, i.e., $\forall \boldsymbol{u}, \boldsymbol{v} \in \mathbb{R}^n$, we have $\mathcal{L}(\boldsymbol{u}) \leq \mathcal{L}(\boldsymbol{v}) + \langle \nabla \mathcal{L}(\boldsymbol{v}), \boldsymbol{u} - \boldsymbol{v} \rangle + \frac{L}{2} \|\boldsymbol{u} - \boldsymbol{v}\|^2.$*

2. *At the outer step $t$ and inner step $p$, $\forall \mathcal{W}_i \in \mathcal{G}_m^r$, $m \in 1, \cdots, M$, the algorithm can access a bounded noisy gradient and the true gradient is bounded, i.e., $\|\boldsymbol{g}_{t,p}^{(i)}\|_\infty \leq G_\infty, \|\mathbb{E}[\boldsymbol{g}_{t,p}^{(i)}]\|_\infty \leq G_\infty, \forall t \in [T-1] := \{0, \cdots, T-1\}, \forall p \in [\tau-1] := \{0, \cdots, \tau-1\}.$*

3. *The noisy gradient is unbiased and the noise is independent, i.e., $\boldsymbol{g}_{t,p}^{(i)} = \mathbb{E}[\boldsymbol{g}_{t,p}^{(i)}] + \boldsymbol{\zeta}_{t,p}^{(i)}, \mathbb{E}[\boldsymbol{\zeta}_{t,p}^{(i)}] = \boldsymbol{0}$ and $\boldsymbol{\zeta}_{t,p}^{(i)}$ is independent of $\boldsymbol{\zeta}_{t',p'}^{(i)}$ if $t \neq t'$ or $p \neq p'$.*

4. *The learning rate of the inner optimizer is $\eta_{t,p} = \eta/\sqrt{t\tau + p + 1}$, and the learning rate of the outer optimizer is $\nu$.*

*Then Algorithm 1 yields*

$$\min_{t \in [T-1], p \in [\tau-1]} \mathbb{E}[\|\nabla \mathcal{L}(\boldsymbol{\theta}_{t,p})\|^2]$$

$$\leq \frac{1}{2\sqrt{\tau}\eta(\sqrt{T}-1)} \left( \frac{\mathcal{L}(\boldsymbol{\theta}_{0,0})}{\nu} + \frac{LnG_\infty^2\tau\phi\eta^2(1+\ln(\tau T))}{\epsilon} + \frac{L\nu nG_\infty^2\phi^2\eta^2(1+\ln(\tau T))}{2\epsilon^2} \right). \quad (6)$$

*where the meaning of $n$, $\phi$ and $\epsilon$ are listed in Table 7 of Appendix A.4.*

The proof of Theorem 1 is presented in Appendix A.4. Therefore, the convergence (to the stationary point) rate of EDiT is $O(\log(T)/\sqrt{T})$.

## 6 CONCLUSION

In this work, we investigate the challenge of training LLMs on large-scale clusters. We analyze the fundamental characteristics of large scale clusters and the limitations of the existing Local SGD-type methods. On this basis, we propose a novel Efficient Distributed Training method for LLMs called EDiT. This method effectively integrates model sharding strategies with tailored Local SGD mechanisms. We propose layer-wise synchronization to achieve overlap of computation and communication and reduce communication and memory overhead. We enhance the convergence and stability of EDiT by introducing a pseudo gradient penalty strategy. We also present an asynchronous variant of EDiT (A-EDiT) to tackle the problem of consistent stragglers in heterogeneous clusters. Extensive experimental results demonstrate the superior capabilities of our proposed methods across multiple dimensions, and the convergence analysis provides a theoretical foundation for our method.

Several potential avenues for future research are identified. First, for the A-EDiT, the stragglers negatively impact the overall performance. Mitigating the impact of these stragglers warrants further investigation. Second, our simulation of elastic training currently entails halting and restarting the training process upon node addition or subtraction. We look forward to a truly elastic framework that can swiftly adjust training resources without disrupting the ongoing training process.

ACKNOWLEDGMENTS

We thank Ke Zhang for bringing the straggler and communication issues in distributed training to our attention and fully supporting our research on this topic. We thank Ji Zhang for providing us with insights into Local SGD. We thank Haitao Zhang for his technical advice and assistance. We thank Chunjie Shen for helping to apply our techniques to practical training scenarios.

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

# A APPENDIX

## A.1 METHOD

Here we provide the formal descriptions of EDiT method and model synchronization with pseudo gradient penalty strategy in Algorithm 1 and Algorithm 2, respectively, to help readers better understand our work.

---

**Algorithm 1** EDiT Algorithm

---

**Require:** $K$ workers $\mathcal{W} = \{\mathcal{W}_1, \cdots, \mathcal{W}_K\}$ with each worker $\mathcal{W}_i$ contains $L$ sharded modules $\boldsymbol{\theta}_{0,0}^{(i,l)} = \boldsymbol{\theta}_0^{(i,l)} = \{\boldsymbol{\theta}_0^{(i,1)}, \cdots, \boldsymbol{\theta}_0^{(i,L)}\}$; $K$ data shards $\mathcal{D} = \{\mathcal{D}_1, \cdots, \mathcal{D}_K\}$; $M \times N$ device mesh with the columns forming model shard groups $\mathcal{G}^s = \{\mathcal{G}_1^s, \cdots, \mathcal{G}_N^s\}$ and rows forming model sync groups $\mathcal{G}^r = \{\mathcal{G}_1^r, \cdots, \mathcal{G}_M^r\}$; outer training steps $T$; sync period $\tau$; warmup steps $t_{warm}$;

1: **for** $t = 0$ to $T - 1$ **do**
2:     **for** $p = 0$ to $\tau - 1$ **do**
3:         **for** worker $\mathcal{W}_i$ **parallel do**
4:             Confirm $\mathcal{W}_i$ is in the model sync group $\mathcal{G}_m^r$ and model shard group $\mathcal{G}_n^s$
5:             $(\boldsymbol{x}_{t,p}^{(i,0)}, \boldsymbol{y}_{t,p}^{(i)}) \sim \mathcal{D}_i$
6:             **for** $l = 1$ to $L$ **do**                         ▷ Forward Pass
7:                 **if** $(t * \tau + p) > t_{warm}$ and $p == 0$ **then**
8:                     Sync parameters in $\mathcal{G}_m^r$: $\boldsymbol{\theta}_{t,0}^{(i,l)} = \boldsymbol{\theta}_t^{(i,l)} = \mathrm{Sync}(\boldsymbol{\theta}_{t-1,\tau}^{(i,l)}; \mathcal{G}_m^r)$
9:                 **end if**
10:                 Gather module parameters in $\mathcal{G}_n^s$: $\boldsymbol{\theta}_{t,p}^{(l)} = \mathrm{AllGather}(\boldsymbol{\theta}_{t,p}^{(i,l)}; \mathcal{G}_n^s)$
11:                 Forward calculation: $\boldsymbol{x}_{t,p}^{(i,l)} = f(\boldsymbol{\theta}_{t,p}^{(l)}, \boldsymbol{x}_{t,p}^{(i,l-1)})$
12:                 Free module parameters: $\boldsymbol{\theta}_{t,p}^{(i,l)} = \mathrm{Shard}(\boldsymbol{\theta}_{t,p}^{(l)})$
13:             **end for**
14:             Calculate loss: $\mathcal{L} = \ell(\boldsymbol{x}_{t,p}^{(i,L)}, \boldsymbol{y}_{t,p}^{(i)})$
15:             **for** $l = L$ to $1$ **do**                     ▷ Backward Pass
16:                  Gather module parameters in $\mathcal{G}_n^s$: $\boldsymbol{\theta}_{t,p}^{(l)} = \mathrm{AllGather}(\boldsymbol{\theta}_{t,p}^{(i,l)}; \mathcal{G}_n^s)$
17:                  Backward calculation: $\widehat{\boldsymbol{g}}_{t,p}^{(l)} = \nabla_{\boldsymbol{\theta}_{t,p}^{(l)}} \mathcal{L}(\boldsymbol{\theta}_{t,p}^{(l)}, \boldsymbol{x}_{t,p}^{(i,l-1)})$
18:                  Sync grads in $\mathcal{G}_n^s$: $\boldsymbol{g}_{t,p}^{(i,l)} = \mathrm{ReduceScatter}(\widehat{\boldsymbol{g}}_{t,p}^{(l)}; \mathcal{G}_n^s)$
19:                  **if** $(t * \tau + p) \leq t_{warm}$ **then**
20:                     Sync grads in $\mathcal{G}_m^r$ (Warmup): $\boldsymbol{g}_{t,p}^{(i,l)} = \mathrm{AllReduce}(\boldsymbol{g}_{t,p}^{(i,l)}; \mathcal{G}_m^r)$
21:                  **end if**
22:                  Free module parameters: $\boldsymbol{\theta}_{t,p}^{(i,l)} = \mathrm{Shard}(\boldsymbol{\theta}_{t,p}^{(l)})$
23:             **end for**
24:         **end for**
25:         **for** module $\boldsymbol{\theta}_{t,p}^{(i,l)}$ **parallel do**             ▷ Local Update
26:             Update parameters: $\boldsymbol{\theta}_{t,p+1}^{(i,l)} = \mathrm{InnerOpt}(\boldsymbol{\theta}_{t,p}^{(i,l)}, \boldsymbol{g}_{t,p}^{(i,l)})$
27:         **end for**
28:     **end for**
29: **end for**

---

---

**Algorithm 2** Sync() in Algorithm 1

---

**Require:** The $m$-th model sync group $\mathcal{G}_m^r = \{\mathcal{W}_1, \cdots, \mathcal{W}_N\}$; the sharded parameters $\boldsymbol{\theta}_{t,\tau}^{(i,l)}$ of the $l$-th module in the $i$-th worker $\mathcal{W}_i$ at outer step $t$ and inner step $\tau$; the corresponding last synced sharded parameters $\boldsymbol{\theta}_t^{(i,l)}$ at outer step $t$;

1: Calculate the pseudo gradient: $\boldsymbol{\Delta}_t^{(i,l)} = \boldsymbol{\theta}_{t,\tau}^{(i,l)} - \boldsymbol{\theta}_t^{(i,l)}$

2: Calculate the pseudo gradient norm: $G_t^{(i,l)} = \|\boldsymbol{\Delta}_t^{(i,l)}\|_2$

3: **if** IsAnomaly$(G_t^{(i,l)})$ **then**                       ▷ Eliminate Anomalies

4:       $G_t^{(i,l)} = \infty$

5: **end if**

6: Sync the pseudo gradient norms: $\gamma_t = \sum_k \exp(-G_t^{(k,l)})$ for $\mathcal{W}_j$ in $\mathcal{G}_m^r$

7: **if** $\gamma == 0$ **then**

8:       Rollback parameters: $\boldsymbol{\theta}_{t+1,0}^{(i,l)} = \boldsymbol{\theta}_t^{(i,l)}$

9: **else**

10:       Calculate the weight: $w_{t,i} = \exp(-G_t^{(i,l)})/\gamma_t$             ▷ Weighted Average

11:       Sync the pseudo gradients: $\bar{\boldsymbol{\Delta}}_t^{(i,l)} = \sum_j w_{t,j} \boldsymbol{\Delta}_t^{(j,l)}$ for $\mathcal{W}_j$ in $\mathcal{G}_m^r$

12:       Clip the pseudo gradient: $\widehat{\boldsymbol{\Delta}}_t^{(i,l)} = \text{Clip}(\bar{\boldsymbol{\Delta}}_t^{(i,l)})$          ▷ Clip Grad, Eq. 2 & 5

13:       Update parameters: $\boldsymbol{\theta}_{t+1}^{(i,l)} = \text{OuterOpt}(\boldsymbol{\theta}_t^{(i,l)}, \widehat{\boldsymbol{\Delta}}_t^{(i,l)})$

14:       Sync parameters: $\boldsymbol{\theta}_{t+1,0}^{i,l} = \boldsymbol{\theta}_{t+1}^{i,l}$

15: **end if**

16: **return** $\boldsymbol{\theta}_{t+1,0}^{(i,l)}$

---

## A.2   Experimental Setups

Here we provide a more detailed description of the experimental setups to facilitate readers in reproducing the experimental results of this paper.

**Models** We consider four different scales of Llama models (Touvron et al., 2023) in our experiments: 350M, 1B, 3B, and 7B. Their specific configurations are detailed in Table 3. We configure the models to have the same number of layers and head dimensions, which facilitates the utilization of $\mu$P (Yang et al., 2021) for hyperparameter search.

Table 3: Configurations for the four scales of Llama models.

| Hyperparameter | 350M | 1B | 3B | 7B |
|---|---|---|---|---|
| Number of Layers | 32 | 32 | 32 | 32 |
| Hidden Size | 768 | 1,536 | 2,560 | 4,096 |
| Intermediate Size | 2,048 | 4,096 | 6,912 | 11,008 |
| Number of Heads | 6 | 12 | 20 | 32 |
| Number of K/V Heads | 6 | 12 | 20 | 32 |
| Vocab Size | 79,800 | | | |

**Datasets** Departing from small language datasets used in prior works (Douillard et al., 2023), we employ a new large-scale open-source dataset, FineWeb-Edu (Lozhkov et al., 2024) in our experiments. This dataset comprises 1.3T tokens of premium educational web pages filtered from the extensive FineWeb repository (Penedo et al., 2024). Additionally, we also utilize an in-house private pre-training dataset, which consists of a diverse collection of corpus of varying quality.

**Baselines** We compare the proposed EDiT and A-EDiT method against several state-of-the-art methods, including standard mini-batch (Baseline), Post Local SGD (Lin et al., 2019), DiLoCo (Douillard et al., 2023), and CO2/CO2* (Sun et al., 2023). Here CO2* is the memory-efficient version of CO2 that shards extra parameters and outer momentum across workers (Sun et al., 2023). Since Parallel SGD (Zhang et al., 2016) and SlowMo (Wang et al., 2019) are equiva-

lent to Post Local SGD (Lin et al., 2019) and DiLoCo (Douillard et al., 2023), respectively, we do not include them in comparisons.

**Training** Follow DiLoCo (Douillard et al., 2023), we use AdamW (Loshchilov & Hutter, 2019) as the inner optimizer and Nesterov momentum (Sutskever et al., 2013) as the outer optimizer. The models are initialized with $\mu$P (Yang et al., 2021), enabling the hyperparameters transfer from the smallest scale model (350M) to models of larger magnitude. To balance efficiency and performance, the synchronization interval $\tau$ and $\tau_{time}$ are set to 128 and 600s, respectively. Across all experiments, a context length of 4,096 tokens and a cosine learning rate decay schedule are consistently applied. For the FineWeb-Edu dataset (Lozhkov et al., 2024), the total batch size is set to 1,024 and the training step is set to 100,000 ($\sim$420B tokens). The learning rate for Baseline, inner learning rate, outer learning rate, and outer momentum are set to 3e-4, 1.5e-4, 0.8, and 0.85, respectively. For the in-house dataset, the total batch size is set to 1,536 and the training step is set to 150,000 ($\sim$950B tokens). The learning rate for Baseline, inner learning rate, outer learning rate, and outer momentum are set to 6e-4, 6e-4, 1.0, and 0.8, respectively. We list the searched hyperparameters in detail in Table 4. The experimental infrastructure comprised eight Nvidia A100 GPU nodes with 64 GPUs and an $8 \times 8$ device mesh. For the hyperparameters in the pseudo gradient penalty strategy, we set $\phi = 10$.

Table 4: The hyperparameters searched in the experiments.

| Hyperparameter | Value | | | | |
|---|---|---|---|---|---|
| Inner Learning Rate | 3e-5 | 6e-5 | 1.5e-4 | 3e-4 | 6e-4 |
| Synchronization Interval | 16 | 64 | 128 | 256 | 512 |
| Outer Learning Rate | 0.5 | 0.7 | 0.8 | 0.9 | 1.0 |
| Outer Momentum | 0.6 | 0.8 | 0.85 | 0.9 | 0.95 |

## A.3 ADDITIONAL EXPERIMENTAL RESULTS

### A.3.1 CONVERGENCE AND GENERALIZATION

In the main text, we present the performance of EDiT and other Local SGD methods in training the Llama 1B model in Figure 4 and Table 1. To demonstrate that EDiT performs consistently well across different model scales, we additionally trained Llama 350M, 3B, and 7B models using EDiT on the FineWeb-Edu dataset, each with a total of 420B tokens. The corresponding training loss, validation PPL, and evaluation results are shown in Figure 8 and Table 5. It can be observed that EDiT is robust across various model scales. Besides, to our knowledge, this is the first time to train a 7B model on a large-scale dataset with a Local SGD-related method. Although CO2 (Sun et al., 2023) also claimed that they trained a 7B model, they only used about 50B tokens and provided only the final validation PPL results.

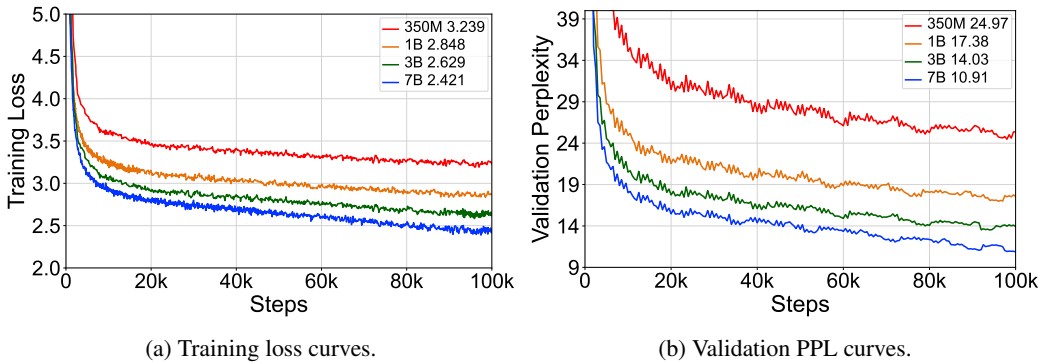

(a) Training loss curves.                    (b) Validation PPL curves.

Figure 8: The training loss and validation PPL curves for the 350M, 1B, 3B, and 7B models trained with the EDiT method on the FineWeb-Edu dataset. The final loss and PPL values are marked in the figures, which are the average values of the last 10 values to prevent randomness.

Table 5: The evaluation results on the public benchmarks (Fourrier et al., 2023; OpenCompass Contributors, 2023) for the 350M, 1B, 3B, and 7B models trained with the EDiT method.

| Benchmark | 350M | 1B | 3B | 7B |
|---|---|---|---|---|
| MMLU (↑) | 28.96 | 32.29 | 34.70 | 36.20 |
| ARC-E (↑) | 53.00 | 60.00 | 67.60 | 68.30 |
| ARC-C (↑) | 25.40 | 32.40 | 36.10 | 39.20 |
| HellaSwag (↑) | 39.44 | 51.75 | 58.55 | 62.31 |
| PIQA (↑) | 65.00 | 68.10 | 72.40 | 73.70 |
| CommonSense-QA (↑) | 29.50 | 36.30 | 37.30 | 42.60 |
| OpenBookQA (↑) | 21.40 | 26.00 | 27.40 | 28.60 |
| WinoGrande (↑) | 50.50 | 51.70 | 52.00 | 52.40 |
| Average (↑) | 39.15 | 44.82 | 48.26 | 50.41 |

### A.3.2 ACCELERATION

In the main text, we analyzed the throughput and TFLOPS of different acceleration methods. Here, we further profile the synchronization operations of different methods when training the Llama 1B model. As shown in Figure 9, Post Local SGD introduces a significant additional communication overhead of 160ms during model synchronization. Although CO2* successfully overlaps model synchronization communication with the forward computation of the next step, it incurs two segments of non-overlapping communication overhead to deal with the sharded extra parameters and outer momentum, causing a delay of approximately 300ms. This delay negates the acceleration benefits gained from overlapped parameter synchronization. However, without the memory-efficient mode, the complete copies of model parameters and outer momentum in CO2 lead to severe memory usage, resulting in OOM in this scenario. In contrast, EDiT synchronizes sharded parameters layer-by-layer during the forward pass, reducing communication volume and overlapping computation with communication through a prefetch strategy. It achieves the same performance as CO2 without introducing additional communication burdens or memory overhead. As a result, EDiT only introduces 19ms delay in this scenario.

Besides, we provide the detailed TFLOPS corresponding to the Figure 5 in Table 6.

Table 6: The TFLOPS of different methods under different training scenarios.

| Random Straggler | | | | Consistent Straggler | | | | Limited Bandwidth | | | |
|---|---|---|---|---|---|---|---|---|---|---|---|
| Lag (s) | Baseline | EDiT | A-EDiT | Lag (s) | Baseline | EDiT | A-EDiT | Repeat | Baseline | EDiT | A-EDiT |
| 0 | 225.75 | 236.50 | 237.45 | 0 | 225.75 | 236.50 | 237.45 | 0 | 225.75 | 236.50 | 237.45 |
| 1.5 | 175.21 | 228.06 | 230.05 | 1.5 | 175.12 | 181.20 | 230.12 | 10 | 205.71 | 234.74 | 237.85 |
| 2.5 | 150.26 | 219.72 | 224.38 | 2.5 | 150.03 | 154.12 | 227.58 | 20 | 136.64 | 236.20 | 238.04 |
| 3.5 | 130.94 | 214.36 | 219.49 | 3.5 | 130.80 | 134.00 | 225.08 | 30 | 105.06 | 236.46 | 237.73 |
| 4.5 | 115.29 | 209.44 | 214.53 | 4.5 | 115.94 | 118.47 | 223.07 | 40 | 85.18 | 236.39 | 238.03 |

### A.3.3 SCALABILITY

Here we provide the detailed training loss curves of the Baseline and EDiT methods under different numbers of workers and distinct learning rates in Figure 10, which correspond to the Figure 6 in the main text.

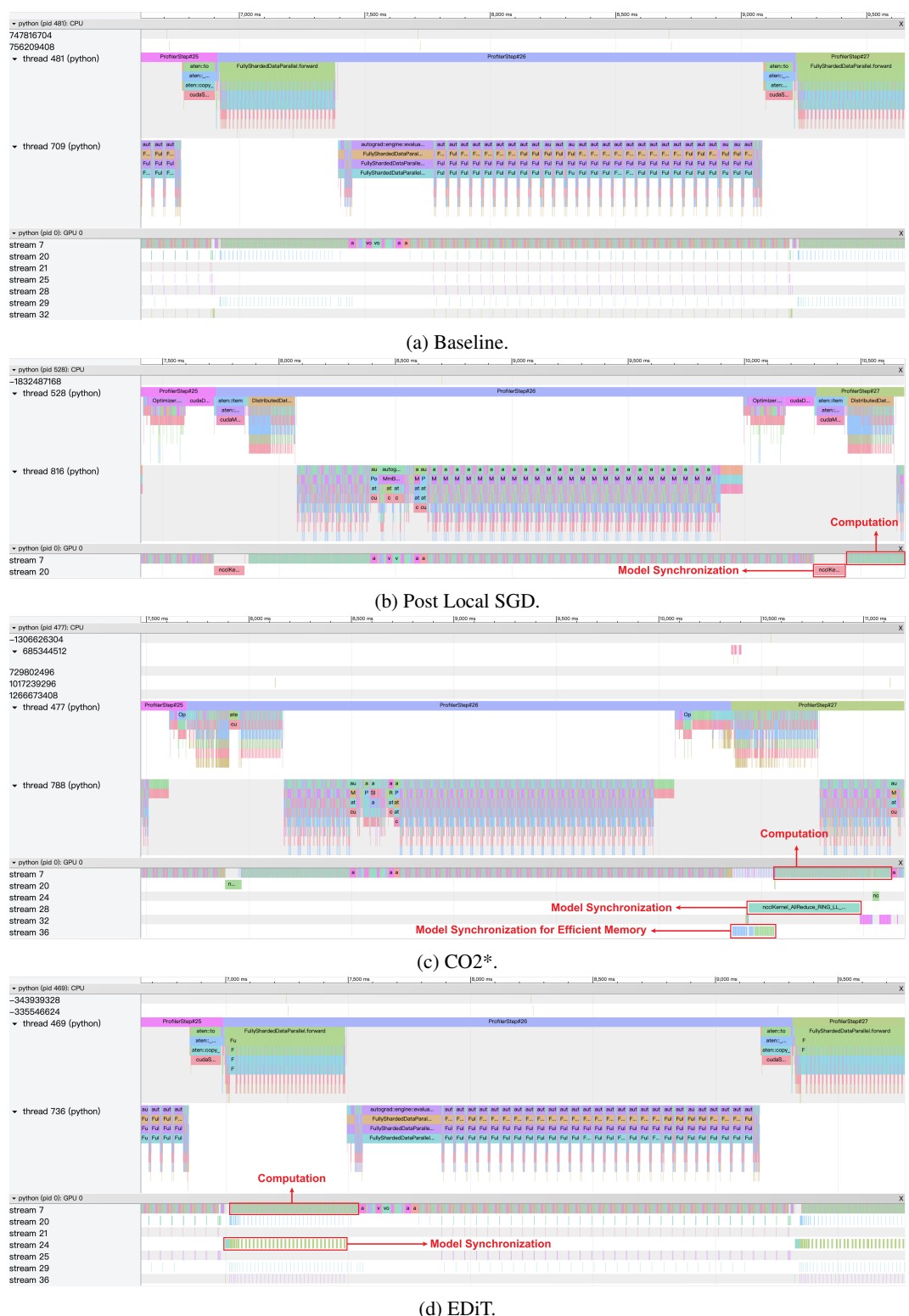

Figure 9: The profiling results of Baseline, Post Local SGD, CO2, and EDiT during synchronization while training the Llama 1B model. The parts corresponding to model synchronization and computation are highlighted with red boxes.

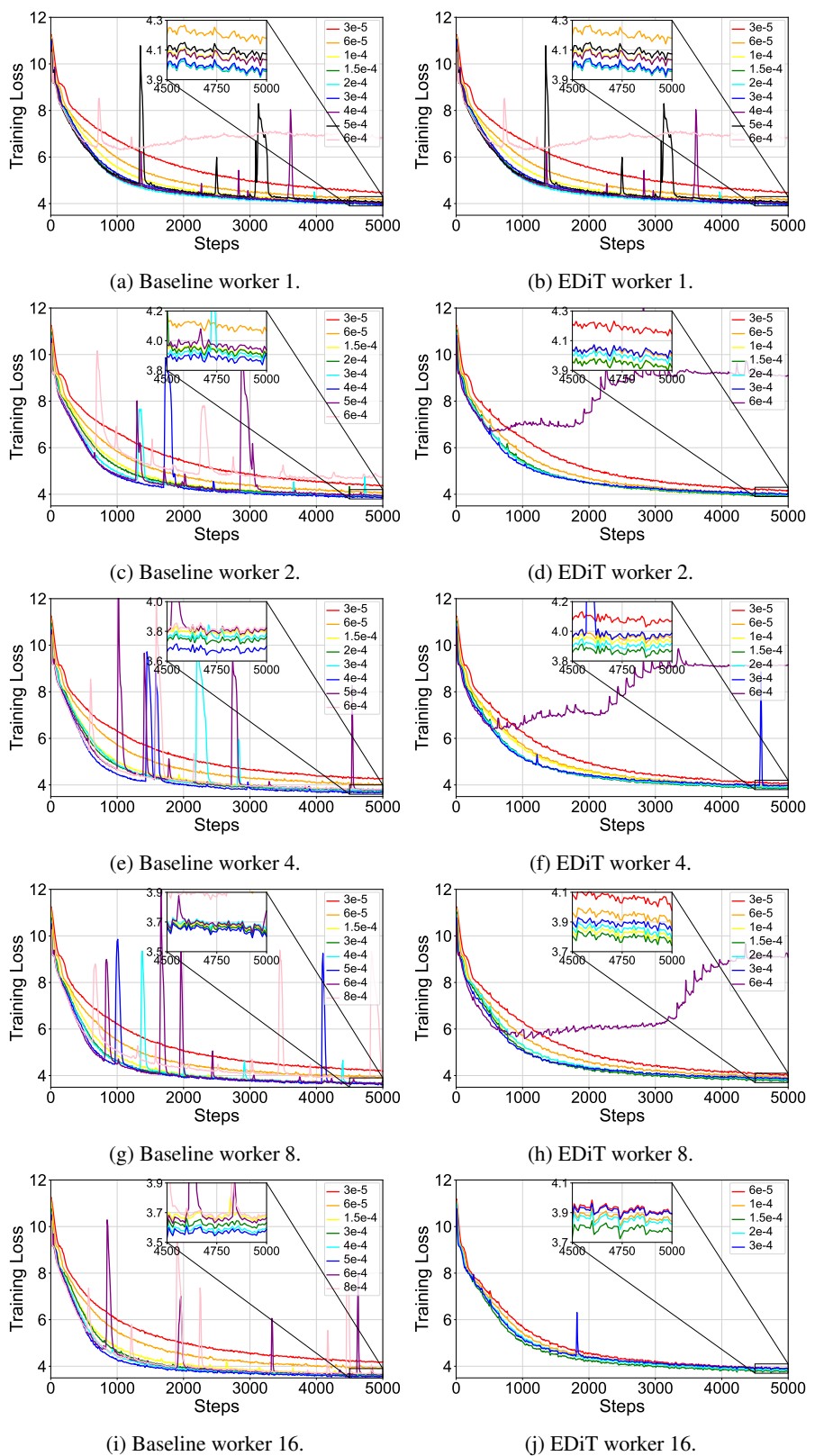

Figure 10: The training loss curves of the Baseline and EDiT methods under different numbers of workers and distinct learning rates.

## A.4 Proof of Theorem 1

Table 7: The hyperparameters of Theorem 1.

| Symbol | Meaning |
|--------|---------|
| $\eta$ | The initial learning rate of the inner optimizer |
| $\nu$ | The learning rate of the outer optimizer |
| $\tau$ | The synchronization interval |
| $\phi$ | The gradient norm clip threshold |
| $L$ | The loss function is $L$-smooth |
| $\epsilon$ | The small positive constant to avoid division by zero |
| $n$ | The dimension of model parameters |
| $G_\infty$ | The upper bound of the gradients |

*Proof.* Since the inner and outer optimizer are both SGD, $\forall\, \mathcal{W}_i \in \mathcal{G}_m^r, m \in \{1, \cdots, M\}$, we have

$$\boldsymbol{\theta}_{t,0}^{(i)} = \boldsymbol{\theta}_t^{(i)}, \tag{7}$$

$$\boldsymbol{\theta}_{t,p+1}^{(i)} = \boldsymbol{\theta}_{t,p}^{(i)} - \eta_{t,p}\boldsymbol{g}_{t,p}^{(i)}, \tag{8}$$

$$\boldsymbol{\theta}_{t+1} = \boldsymbol{\theta}_t - \nu\widehat{\boldsymbol{\Delta}}_t, \tag{9}$$

where $\eta_{t,p}$ and $\nu$ are the inner optimizer learning rate and the outer optimizer learning rate, respectively. Here, we omit the superscripts of variables in Equation 9, as they remain identical across all workers in $\mathcal{G}_m^r$. The following proof will also adopt this simplified notation without risk of confusion. Hence, by the Equations 2, 3, 4, 5, 7 and 8, we have

$$\widehat{\boldsymbol{\Delta}}_t = \beta_t\bar{\boldsymbol{\Delta}}_t = \beta_t \sum_j w_{t,j}\boldsymbol{\Delta}_t^{(j)} = \beta_t \sum_j w_{t,j}(\boldsymbol{\theta}_{t,\tau}^{(j)} - \boldsymbol{\theta}_{t,0}^{(j)})$$

$$= \beta_t \sum_j w_{t,j}(\eta_{t,p}\boldsymbol{g}_{t,\tau-1}^{(j)} + \boldsymbol{\theta}_{t-1,\tau}^{(j)} - \boldsymbol{\theta}_{t,0}^{(j)}) = \beta_t \sum_j w_{t,j} \sum_{p=0}^{\tau-1} \eta_{t,p}\boldsymbol{g}_{t,p}^{(j)}. \tag{10}$$

Let

$$\boldsymbol{h}_{t,p} = \beta_t \sum_j w_{t,j}\boldsymbol{g}_{t,p}^{(j)}, \tag{11}$$

then

$$\mathbb{E}[\boldsymbol{h}_{t,p}] = \mathbb{E}[\beta_t \sum_j w_{t,j}\boldsymbol{g}_{t,p}^{(j)}] = \beta_t\mathbb{E}[\boldsymbol{g}_{t,p}]. \tag{12}$$

Combining Equation 7, Equation 10 and Equation 11 into Equation 9, we have

$$\boldsymbol{\theta}_{t+1,0} - \boldsymbol{\theta}_{t,0} = -\nu \sum_{p=0}^{\tau-1} \eta_{t,p}\boldsymbol{h}_{t,p}. \tag{13}$$

For proving the convergence of $\{\boldsymbol{\theta}_{t,p}^{(i)}\}$, we need to define the auxiliary sequence $\{\boldsymbol{\psi}_{t,p}\}$. Denote

$$\begin{cases} \boldsymbol{\psi}_{0,0} = \boldsymbol{\theta}_{0,0}, \\ \boldsymbol{\psi}_{t+1,0} = \boldsymbol{\psi}_{t,0} - \nu\sum_{p=0}^{\tau-1}\eta_{t,p}\boldsymbol{h}_{t,p}, \\ \boldsymbol{\psi}_{t,p+1} = \boldsymbol{\psi}_{t,p} - \nu\eta_{t,p}\boldsymbol{h}_{t,p}. \end{cases}$$

It is easy to prove $\boldsymbol{\psi}_{t+1,0} = \boldsymbol{\psi}_{t,\tau}$. Then we have

$$\mathbb{E}[\boldsymbol{\psi}_{t,p}] - \mathbb{E}[\boldsymbol{\theta}_{t,p}] = \mathbb{E}[\boldsymbol{\psi}_{t,0} - \nu\sum_{k=0}^{p-1}\eta_{t,k}\boldsymbol{h}_{t,k}] - \mathbb{E}[\boldsymbol{\theta}_{t,0} - \sum_{k=0}^{p-1}\eta_{t,k}\boldsymbol{g}_{t,k}] = (1 - \nu\beta_t)\sum_{k=0}^{p-1}\eta_{t,k}\mathbb{E}[\boldsymbol{g}_{t,k}]$$

$$\tag{14}$$

By assumption 1, we have

$$\mathcal{L}(\boldsymbol{\psi}_{t,p+1}) \leq \mathcal{L}(\boldsymbol{\psi}_{t,p}) + \langle \nabla \mathcal{L}(\boldsymbol{\psi}_{t,p}), \boldsymbol{\psi}_{t,p+1} - \boldsymbol{\psi}_{t,p} \rangle + \frac{L}{2} \|\boldsymbol{\psi}_{t,p+1} - \boldsymbol{\psi}_{t,p}\|^2$$

$$= \mathcal{L}(\boldsymbol{\psi}_{t,p}) - \nu \eta_{t,p} \langle \nabla \mathcal{L}(\boldsymbol{\psi}_{t,p}), \boldsymbol{h}_{t,p} \rangle + \frac{L\nu^2 \eta_{t,p}^2}{2} \|\boldsymbol{h}_{t,p}\|^2$$

$$= \mathcal{L}(\boldsymbol{\psi}_{t,p}) - \nu \eta_{t,p} \left\langle \nabla \mathcal{L}(\boldsymbol{\psi}_{t,p}) - \nabla \mathcal{L}(\boldsymbol{\theta}_{t,p}^{(i)}), \boldsymbol{h}_{t,p} \right\rangle - \nu \eta_{t,p} \left\langle \nabla \mathcal{L}(\boldsymbol{\theta}_{t,p}^{(i)}), \boldsymbol{h}_{t,p} \right\rangle + \frac{L\nu^2 \eta_{t,p}^2}{2} \|\boldsymbol{h}_{t,p}\|^2$$

$$\leq \mathcal{L}(\boldsymbol{\psi}_{t,p}) + \nu \eta_{t,p} L \|\boldsymbol{\psi}_{t,p} - \boldsymbol{\theta}_{t,p}^{(i)}\| \|\boldsymbol{h}_{t,p}\| - \nu \eta_{t,p} \left\langle \nabla \mathcal{L}(\boldsymbol{\theta}_{t,p}^{(i)}), \boldsymbol{h}_{t,p} \right\rangle + \frac{L\nu^2 \eta_{t,p}^2}{2} \|\boldsymbol{h}_{t,p}\|^2. \tag{15}$$

Rearranging Equation 15 and taking expectation both sides, by assumption 2, assumption 3, Equation 12 and Equation 14, we get

$$\nu \eta_{t,p} \beta_t \mathbb{E}[\|\nabla \mathcal{L}(\boldsymbol{\theta}_{t,p})\|^2]$$

$$\leq \mathbb{E}[\mathcal{L}(\boldsymbol{\psi}_{t,p}) - \mathcal{L}(\boldsymbol{\psi}_{t,p+1})] + \nu \eta_{t,p} L \mathbb{E}[\|\boldsymbol{\psi}_{t,p} - \boldsymbol{\theta}_{t,p}\| \|\boldsymbol{h}_{t,p}\|] + \frac{L\nu^2 \eta_{t,p}^2}{2} \mathbb{E}[\|\boldsymbol{h}_{t,p}\|^2]$$

$$\leq \mathbb{E}[\mathcal{L}(\boldsymbol{\psi}_{t,p}) - \mathcal{L}(\boldsymbol{\psi}_{t,p+1})] + \nu \eta_{t,p} L \beta_t \sqrt{n} G_\infty \mathbb{E}[\|\boldsymbol{\psi}_{t,p} - \boldsymbol{\theta}_{t,p}\|] + \frac{L\nu^2 \eta_{t,p}^2 \beta_t^2 n G_\infty^2}{2}$$

$$= \mathbb{E}[\mathcal{L}(\boldsymbol{\psi}_{t,p}) - \mathcal{L}(\boldsymbol{\psi}_{t,p+1})] + \nu \eta_{t,p} L \beta_t \sqrt{n} G_\infty (1 - \nu \beta_t) \sum_{k=0}^{p-1} \eta_{t,k} \mathbb{E}[\|\boldsymbol{g}_{t,k}\|] + \frac{L\nu^2 \eta_{t,p}^2 \beta_t^2 n G_\infty^2}{2}$$

$$\leq \mathbb{E}[\mathcal{L}(\boldsymbol{\psi}_{t,p}) - \mathcal{L}(\boldsymbol{\psi}_{t,p+1})] + \nu \eta_{t,0}^2 L \beta_t n G_\infty^2 \tau + \frac{L\nu^2 \eta_{t,p}^2 \beta_t^2 n G_\infty^2}{2}. \tag{16}$$

Telescoping Equation 16 for $p = 0$ to $\tau - 1$ and $t = 0$ to $T - 1$, we have

$$\sum_{t=0}^{T-1} \sum_{p=0}^{\tau-1} \nu \eta_{t,p} \beta_t \mathbb{E}[\|\nabla \mathcal{L}(\boldsymbol{\theta}_{t,p})\|^2]$$

$$\leq \mathbb{E}[\mathcal{L}(\boldsymbol{\psi}_{0,0}) - \mathcal{L}(\boldsymbol{\psi}_{T-1,\tau})] + \nu L n G_\infty^2 \tau^2 \sum_{t=0}^{T-1} \beta_t \eta_{t,0}^2 + \frac{L\nu^2 n G_\infty^2}{2} \sum_{t=0}^{T-1} \sum_{p=0}^{\tau-1} \beta_t^2 \eta_{t,p}^2 \tag{17}$$

$$\leq \mathcal{L}(\boldsymbol{\theta}_{0,0}) + \nu L n G_\infty^2 \tau^2 \sum_{t=0}^{T-1} \beta_t \eta_{t,0}^2 + \frac{L\nu^2 n G_\infty^2}{2} \sum_{t=0}^{T-1} \sum_{p=0}^{\tau-1} \beta_t^2 \eta_{t,p}^2.$$

Since from Equation 4, we have $1 \leq \beta_t \leq \frac{\phi}{\epsilon}$. Combining with Equation 17, we can get

$$\sum_{t=0}^{T-1} \sum_{p=0}^{\tau-1} \eta_{t,p} \mathbb{E}[\|\nabla \mathcal{L}(\boldsymbol{\theta}_{t,p})\|^2] \leq \frac{\mathcal{L}(\boldsymbol{\theta}_{0,0})}{\nu} + \frac{L n G_\infty^2 \tau^2 \phi}{\epsilon} \sum_{t=0}^{T-1} \eta_{t,0}^2 + \frac{L\nu n G_\infty^2 \phi^2}{2\epsilon^2} \sum_{t=0}^{T-1} \sum_{p=0}^{\tau-1} \eta_{t,p}^2. \tag{18}$$

Since

$$\sum_{t=0}^{T-1} \sum_{p=0}^{\tau-1} \eta_{t,p} \geq \tau \sum_{t=0}^{T-1} \eta_{t,\tau-1} = \sqrt{\tau} \eta \sum_{t=0}^{T-1} \frac{1}{\sqrt{t+1}}$$

$$= \sqrt{\tau} \eta \left( \int_1^2 \frac{1}{\sqrt{1}} ds + \cdots + \int_T^{T+1} \frac{1}{\sqrt{T}} ds \right) \geq \sqrt{\tau} \eta \int_1^{T+1} \frac{1}{\sqrt{s}} ds$$

$$= 2\sqrt{\tau} \eta (\sqrt{T+1} - 1) \geq 2\sqrt{\tau} \eta (\sqrt{T} - 1),$$

$$\sum_{t=0}^{T-1} \sum_{p=0}^{\tau-1} \eta_{t,p}^2 \leq \tau \sum_{t=0}^{T-1} \eta_{t,0}^2 = \eta^2 \sum_{t=0}^{T-1} \frac{1}{t + \frac{1}{\tau}} = \eta^2 \left( 1 + \int_0^1 \frac{1}{1 + \frac{1}{\tau}} ds + \cdots + \int_{T-2}^{T-1} \frac{1}{T - 1 + \frac{1}{\tau}} ds \right)$$

$$\leq \eta^2 \left( 1 + \int_0^{T-1} \frac{1}{s + \frac{1}{\tau}} ds \right) = \eta^2 (1 + \ln(\tau T - \tau + 1)) \leq \eta^2 (1 + \ln(\tau T)), \tag{19}$$

substituting Equation 19 into Equation 17, we have

$$\min_{t\in[T-1],p\in[\tau-1]} \mathbb{E}[\|\nabla\mathcal{L}(\boldsymbol{\theta}_{t,p})\|^2]$$

$$\leq \frac{1}{\sum_{t=0}^{T-1}\sum_{p=0}^{\tau-1}\eta_{t,p}} \left( \frac{\mathcal{L}(\boldsymbol{\theta}_{0,0})}{\nu} + \frac{LnG_\infty^2\tau^2\phi}{\epsilon}\sum_{t=0}^{T-1}\eta_{t,0}^2 + \frac{L\nu nG_\infty^2\phi^2}{2\epsilon^2}\sum_{t=0}^{T-1}\sum_{p=0}^{\tau-1}\eta_{t,p}^2 \right)$$

$$\leq \frac{1}{2\sqrt{\tau}\eta(\sqrt{T}-1)} \left( \frac{\mathcal{L}(\boldsymbol{\theta}_{0,0})}{\nu} + \frac{LnG_\infty^2\tau\phi\eta^2(1+\ln(\tau T))}{\epsilon} + \frac{L\nu nG_\infty^2\phi^2\eta^2(1+\ln(\tau T))}{2\epsilon^2} \right).$$

$$(20)$$

This completes the proof. $\qquad\square$

