# OpenReview forum: "EDiT: A Local-SGD-Based Efficient Distributed Training Method for Large Language Models"
_ICLR.cc/2025/Conference — ICLR 2025 Poster_

### Official Review · Reviewer_eSB4 · 2024-11-05

**Soundness:** 2
**Presentation:** 2
**Contribution:** 2
**Rating:** 6
**Confidence:** 4

**Summary:**

This paper proposed a distributed training approach which arranges the participating workers into a two-dimensional device mesh, i.e., model replica group and the model shard group. It also introduces a pseudo gradient penalty mechanism that eliminates the significantly anomalous worker and averages and clips the gradients. It also introduces an asynchronous variant of the proposed scheme. Experiments have been conducted to evaluate the performance of the proposed scheme on the distributed training of LLMs.

**Strengths:**

It is interesting and novel idea of arranging the participating workers into a two-dimensional device mesh, i.e., model replica group and the model shard group. The experimental results also demonstrate the effectiveness of the proposed pseudo gradient penalty mechanism and asynchronous version of the proposed scheme. The presentation of this paper is fair and easy to follow.

**Weaknesses:**

1. The novelty of this paper is limited. The author claimed that this paper is the first to integrate Local SGD with modern distributed strategies. What does "modern distributed strategies" refers to here is not clear. Is't local SGD a modern distributed strategy. The way to eliminate the significantly anomalous worker and averages and clips the gradients is not a new idea.
2. The  hierarchical distributed training method on a 2-D device mesh may be problematic. Since each worker within model shard group runs a subpart of the model, it may need to wait for the backward gradients of the same batch before processing the next batch, leading to significant idle time and slowing down the training.
3. The improvement over the existing work since quite insignificant as shown in Figure 4. The performance of the proposed scheme is only compared to baseline, which is not sufficient to demonstrate its superiority. The author should compare it with a more advance asynchronous scheme, such as following work:
[1] Nguyen J, Malik K, Zhan H, et al. Federated learning with buffered asynchronous aggregation[C]//International Conference on Artificial Intelligence and Statistics. PMLR, 2022: 3581-3607.

**Questions:**

1. The proposed hierarchical distributed training method on a 2-D device mesh should be better explained, like how workers within a model shard group coordinate with each other. What are the messages transmit between them (should be intermediate results instead of parameters?)
2. Is there any design in the proposed method addressing the concern that each worker within model shard group may need to wait for the backward gradients of each batch from the next worker in the same group?
3. How this proposed hierarchical distributed training method accelerates the distributed training? It should be evaluated in the numerical results.
4. In Section 4.5, it seems suggesting that DiLoCo scheme with the pseudo gradient penalty mechanism is the proposed scheme EDiT?

---

> ### Author Response · Authors · 2024-11-26
> **Response to Reviewer eSB4 (Part 1/2)**
>
> Dear Reviewer,
>
> We appreciate your thorough evaluation of our paper. Below, we address your concerns and provide our responses. All revisions will be included in the forthcoming rebuttal version of the paper.
>
> **Q1: The novelty of this paper is limited.**
>
> A1: Thank you very much for your questions. Our responses are as follows:
> 1. The "modern distributed strategies" mentioned here refer to strategies such as Fully Sharded Data Parallel (FSDP), ZeRO, or 3D parallelism used for training LLMs. We also realized that the term "modern distributed strategies" is vague. Therefore, we have revised and specified it as "model sharding strategy," which refers to that shard model parameters to prevent out of memory. This terminology is more appropriate for the methods discussed in our paper and avoids potential confusion.
> 2. Although eliminating significant anomalous workers, averaging and clipping gradients are not new to deep learning community, to our knowledge, we are the first to propose using these strategies within the context of Local SGD to mitigate training instability caused by highly diverse data and worker variability. These strategies integrate elegantly with the Local SGD method, and we also provide experimental evidence to show that they are necessary when training LLMs with complex data.
>
> In this paper, we introduce EDiT, an innovative efficient distributed method that combines a tailored Local SGD approach with model sharding techniques to enhance large-scale training efficiency. EDiT performs layer-wise parameter synchronization during forward pass and employs a prefetch strategy to overlap computation and communication, thereby reducing the additional communication and GPU memory overhead introduced by parameter synchronization. Besides, EDiT employs a novel pseudo gradient penalty strategy to suppress loss spikes caused by suboptimal corpus across workers, which ensures training stability and improve performance. Additionally, we propose an asynchronous variant of the EDiT method named A-EDiT to deal with the consistent stragglers in heterogeneous clusters. We also provide a large-scale verification of asynchronous pre-training for LLMs, along with an extensive analysis of convergence, generalization, acceleration, scalability, and stability. Based on these points, we believe that our work is indeed novel.
>
> **Q2: It may need to wait for the backward gradients, leading to significant idle time and slowing down the training.**
>
> A2: Thank you for your question. In the model shard group, the model is sharded horizontally rather than vertically (i.e., pipeline parallelism). Therefore, there is no issue of one worker needing to wait for another worker to return gradients. All workers in a model shard group compute the gradients for a given layer simultaneously.
>
> **Q3: The improvement over the existing work since quite insignificant as shown in Figure 4.**
>
> A3: Thank you very much for your question.
> 1. In our experiments, we compared EDiT with the Baseline and a series of classical or state-of-the-art methods, including Post Local SGD, DiLoCo, and CO2. It is worth noting that most methods that introduce periodic synchronization to improve training throughput has the problem of model performance decline. However, our proposed EDiT method not only improves training throughput but also achieves training and testing results that are consistent with or even better than the Baseline, which convincingly demonstrates the effectiveness of our approach.
> 2. Regarding the work [Federated learning with buffered asynchronous aggregation](https://arxiv.org/abs/2106.06639) that you mentioned, we have cited it in the related work section. This work is based on a Parameter-Server architecture for federated learning that is not suitable for training LLMs. EDiT and other compared methods all adopt an All-Reduce architecture. The compared methods in our work are consistent with the state-of-the-art work CO2/DiLoCo.
>
> **Q4: No theoretical results on the convergence of the proposed approach.**
>
> A4: Thank you very much for your constructive suggestion. We have added the convergence analysis for our proposed EDiT method. For details, please refer to "Section 5: Theoretical Analysis" in the main text and "Section A.4: Proof of Theorem 1" in the appendix.

---

> ### Author Response · Authors · 2024-11-26
> **Response to Reviewer eSB4 (Part 2/2)**
>
> **Q5: The proposed hierarchical distributed training method on a 2-D device mesh should be better explained.**
>
> A5:  The 2-D device mesh is composed of model shard groups and model sync groups, which can be understood as the row and column dimensions, respectively.
> 1. In each model shard group, model parameters are evenly sharded among all workers. All-gather communication is performed among these workers to aggregate the parameters before forward and backward computations. After the backward computation, reduce-scatter communication is conducted to synchronize gradients among these workers.
> 2. In each model synchronization group, all workers maintain the same parameter shards and periodically synchronize parameters. This involves performing All-Reduce communication to average the parameters, followed by computing pseudo gradients and updating parameters with the outer optimizer.
>
> Due to the frequent communication required in model shard groups, efficient communication links are necessary. Normally, all workers within a single node form a communication shard group. In contrast, model sync groups require less frequent communication, so high-efficiency communication links are not essential. Typically, all workers with the same rank across different nodes are gathered to form a model sync group. We have updated Figure 1, Section 1 Paragraph 3, Section 3.1, and Algorithm 1 in the main text for a better presentation of the idea and we hope this would remove the confusion you have.
>
> **Q6: Is there any design in the proposed method addressing the concern that each worker within model shard group may need to wait for the backward gradients of each batch from the next worker in the same group?**
>
> A6:  Referring to the response A2: In the model shard group, the model is sharded horizontally rather than vertically (i.e., pipeline parallelism). Therefore, there is no issue of one worker needing to wait for another worker to return gradients. All workers compute the gradients for a given layer simultaneously.
>
> **Q7: How this proposed hierarchical distributed training method accelerates the distributed training? It should be evaluated in the numerical results.**
>
> A7: Thank you for pointing this out. The hierarchical distributed training method we propose leverages the efficient communication links within the model shard groups and the low communication frequency within the model sync groups to achieve training acceleration. Besides, the proposed layer-wise sync method further reduces the time of model synchronization. In Section 4.3, we have added a speed comparison for different methods, and the numerical results demonstrate the acceleration effect of our method.
>
> **Q8: In Section 4.5, it seems suggesting that DiLoCo scheme with the pseudo gradient penalty mechanism is the proposed scheme EDiT?**
>
> A8: Thank you for pointing this out. Indeed, we incorporated a pseudo gradient penalty strategy into our method to enhance training stability and model performance. However, our main contribution lies in integrating Local SGD with model sharding strategy for training LLMs. Unlike DiLoCo and other methods that globally synchronize all parameters after the optimizer updates, EDiT synchronizes the sharded parameters layer by layer during the forward pass, addressing the memory and communication issues present in previous methods.
>
> To avoid any misunderstanding, we have revised the statement in Section 4.5 to "EDiT w/o ALL." In fact, our ablation experiments were indeed conducted based on the EDiT framework.

---

### Official Review · Reviewer_zyrD · 2024-11-05

**Soundness:** 3
**Presentation:** 2
**Contribution:** 2
**Rating:** 6
**Confidence:** 4

**Summary:**

This paper introduces EDiT, a distributed training method leveraging local SGD to reduce communication overhead when training LLMs. The method relies on a "2D-grid" graph topology with a total of $K= M \times N$ data parallel models. On highly connected GPUs (e.g., on the same compute node in a cluster), $N$ models are sharded and communicate at each optimizer step. On the other hand, $M$ of these islands of highly connected machines are connected through possibly lower bandwidth communication links (e.g., on separate compute nodes), and leverage the lower communication frequency needs of local SGD to alleviate communication bottlenecks. To counter the destabilizing effect of using lower batch sizes in local SGD, a gradient penalty method is introduced. Finally, an asynchronous version of the method is also introduced to mitigate the effect of stragglers.

**Strengths:**

* **The novel gradient penalty method seems effective**: In the experiment on the in-house dataset (Fig 4.d), the unstable behavior of standard local-SGD methods is highlighted, and the gradient penalty introduced seems to be an effective solution to alleviate it.
* **Advantages of a proper distributed implementation of a local SGD algorithm are highlighted**: Section 4.3 displays the practical advantages (in terms of throughput) given by a proper implementation of local SGD algorithms using modern distributed methods (such as ZeRO-3/FSDP) when communication links between distributed workers have limited bandwidth.
* **Asynchronous extension**: In the presence of stragglers, the advantage (in terms of throughput) of using an asynchronous extension of EDiT are highlighted in Sec.4.3.

**Weaknesses:**

* **Claiming novelty on the distributed implementation of a local SGD algorithm seems too strong:** As the sharding strategy itself cannot be claimed as novel (see the ZeRO series, FSDP), the “distributed implementation” cannot either. In fact, apart from the gradient penalty method, the Algorithm seems to be a straightforward and natural way of implementing local SGD for LLM training (using ZeRO-3) and not particularly novel (previous local SGD papers such as CO2 [[Sun et al., 2024]](https://arxiv.org/pdf/2401.16265) seem to already make use of ZeRO for their Autoregressive Language Modeling task, and SlowMo [[Wang et al., 2020]](https://arxiv.org/pdf/1910.00643 ) already consider all GPUs inside a cluster node to be a single local worker in their experiments so that each local worker reaches larger batch sizes through data-parallel). Thus, although having a *detailed* pseudo-code for this implementation is welcome, making it the main contribution of the paper and dedicating so much space for explaining it weakens the paper for me. For instance, as the stabilizing effect of the gradient penalty is empirically demonstrated, focusing on this novel observation and contribution seems more relevant to me.
* **No comparison with CO2**: while experiments with SlowMo/DiLoCo are performed, no comparison with state-of-the art method CO2  [[Sun et al., 2024]](https://arxiv.org/pdf/2401.16265) is done.
* **Hyper-parameters introduced:** EDiT introduces 3 novel hyper-parameters $\alpha, \delta, \phi$, but the impact of their values on the method is not discussed.
* **Lack of clarity and imprecision in the writing:**
  * It was not clear at first that the $N$ models were *data parallel sharded* model replicas and **not** model-parallel splits. Maybe clarifying this distinction early in the paper could avoid potential confusion.
  * **line 21:** *“ensure training stability and improve performance.”* this is a bit cryptic. Maybe detailing the reason for these instabilities (as done lines 210-213) early in the paper would help understand the challenges tackled here more clearly.
  * **line 50:** *“Current Local SGD methods do not integrate well with modern distributed strategies”*. Why? Can you explain your point? For instance, the CO2 paper claims that their algorithm is compatible with ZeRO-series optimizers.
  * **lines 73-75:** the problem described is exactly the one the CO2 paper aims at solving, so saying *“current local-SGD method”* seems to be an overstatement here.
  * **line 116:** *“DiLoCo extends the slow momentum in SlowMo to the outer optimizer.”* this statement does not seem accurate: the momentum in SlowMo is already in the outer-loop, and seems to be roughly equivalent to DiLoCo (with nesterov SGD as the outer optimizer, which is also noted line 752). Can you clarify in which way DiLoCo extends SlowMo in this context?
  * **line 170:** *“model synchronization”*. What does this mean? Can you provide a clear definition or explanation of this?

**Questions:**

* **Intra node for model shards:** on standard GPU clusters using nodes with 8 A100 or H100 GPUs with 80GB of memory, what size of model can be loaded & how many intra-node model replicas does it lead to?
* **line 168:** a warm up phase is used as in post-local SGD. However, DiLoCo finds that it is not really helpful (cf their Fig.3), do you observe a different behavior here?
* **Tab.1 & Fig.4**: Why not experiment with A-EDiT on the in-house dataset?
* **Fig 5.a)**: can you explain the reason why A-EDiT and EDiT exhibit the same throughput in the random-straggler scenario?
* **In ablation Fig 7.a):** the Gradient Clipping doesn’t seem to have much effect in the validation PPL curve, is it really necessary to keep it along with WA and anomaly detection (which both seem to smooth the spikes observed in the validation PPL curve during training)?

=========REBUTTAL==========

The authors correctly addressed my concerns and provided further precision on their method, I raise my score.

---

> ### Author Response · Authors · 2024-11-26
> **Response to Reviewer zyrD (Part 1/4)**
>
> Dear Reviewer,
>
> Thank you for your insightful and constructive feedback, which has significantly contributed to the improvement of our paper. Bellow, we address your concerns in detail. All the revisions will be included in the forthcoming rebuttal version of the paper.
>
> **Q1: Claiming novelty on the distributed implementation of a local SGD algorithm seems to strong.**
>
> A1 (Part 1): Thank you very much for your helpful comment. As you mentioned, the sharding strategies and implementations of the ZeRO series/FSDP themselves are not novel. However, combining them with Local SGD methods, especially effectively integrating model sharding into Local SGD, had not been explicitly explored prior to our work. Moreover, EDiT is a method specifically designed for these distributed training frameworks (ZeRO/FSDP), leveraging the characteristics of both Local SGD and the distributed frameworks, and thus go beyond a straightforward combination of ZeRO and LocalSGD. Specifically, most previous Local SGD methods synchronize all parameters globally after optimizer updates, leading to remarkable communication overhead. Additionally, when using outer momentum, previous methods introduce extra memory usage (twice the size of the model), and CO2, due to its one-step-delay strategy, incurs even greater memory usage (four times the size of the model), making them less preferrable to scale to larger LLMs. In contrast, EDiT first proposes layer-wise synchronization of sharded parameters during forward computation, achieving the following advantages:
>
> 1. Significantly reduced communication and data transfer burden. For instance, when training the 32-layer 1B model in the paper on 8 A100 GPUs, EDiT's communication volume per step is $1/(8 \times 32)$ of other methods. Furthermore, EDiT only requires communication within each model sync group, making it more efficient than previous methods' global communication.
> 2. Greatly reduced extra memory overhead. In EDiT, each GPU only needs to store sharded extra parameters and outer momentum, which can be further saved to CPU memory, incurring no additional memory overhead. In the layer-by-layer synchronization mode, CPU-GPU data transfer can also be overlapped with computation.
> 3. The prefetch strategy naturally achieves overlap of computation and communication. Using prefetch, EDiT can synchronize and aggregate the next layer’s parameters while computing the current layer, thus overlapping computation with communication and GPU-CPU data transfer.
>
> We have supplemented speed comparison experiments in the paper. As shown in Table 2 and Figure 9, EDiT achieves throughput comparable to CO2 while handling larger models. The profiling results in Figure 9 also demonstrate that EDiT can achieve nearly 100% overlap of computation and communication when training the 1B model. EDiT better utilizes the characteristics of current distributed frameworks, providing a new solution for model synchronization, which we believe is novel.

---

> ### Author Response · Authors · 2024-11-26
> **Response to Reviewer zyrD (Part 2/4)**
>
> A1 (Part 2): Regarding SlowMo and CO2, as you pointed out, both mention the use of advanced distributed frameworks, and CO2 specifically mentions integrating ZeRO series optimizers to adapt to large model training. However, with detailed study of their papers, codes, and author responses, we found the following issues:
> 1. The open-source codes of CO2 are primarily modified based on the open-source SlowMo codes. First, both of them are variants to FairScale's DataParallelDistributed (DDP) implementation, where each GPU stores the complete gradients, optimizer states, and model parameters, facing large memory burden. The extra parameters and outer momentum further increase memory pressure, making them unsuitable for training LLMs. Although they provide a memory-efficient option, this operation can only shards extra parameters and outer momentum, and does not address the memory usage of the model, optimizer, and gradients. Meanwhile, this operation introduces extra communication overhead. EDiT’s 2D device mesh combines an efficient model sharding strategy with low-frequency communication method to take advantage of the interconnect topology. This addresses the problem of memory/communication efficiency and enlarge the local worker batch size at the same time. While methods like SlowMo/CO2's hierarchical local sgd variants can only solve the latter problem.
> 2. We failed to find any open sourced implementation of the aclaimed CO2-ZeRO integration. From our perspective, CO2 is not readily compatible with ZeRO. In the cases of ZeRO-1 and ZeRO-2, the gradients/optimizer states are sharded, with each GPU responsible for updating only its corresponding portion of parameters. If the parameters are synchronized after each internal update, the overall training process degenerates into a synchronized mode, in which case CO2 will only save the gradient synchronization communication and lose the advantage of Local SGD in alleviating issues such as stragglers. However, if the model is not synchronized after each internal update, meaning each GPU updates the corresponding parameters based only on local sharded gradients, it may lead to unstable training. In the case of ZeRO-3, the authors of CO2 mentioned that in an [OpenReview response](https://openreview.net/forum?id=ZO5cn4IfaN&noteId=ynHafwj2OM): "As a result, communication operations for CO2 with ZeRO-3 involve only two all-gather collective communication primitives at the beginning of the forward and backward passes, respectively, which differs from the original implementation of ZeRO-3." In this scenario, the two all-gather operations introduced by CO2 also cause training to degenerate into a synchronized mode, eliminating the performance gain.
>
> In summary, we believe that existing methods like SlowMo and CO2 are not designed for large scale distributed training frameworks and thus do not integrate well with them, while our method offers significant improvements. Thus in the main text we have put a detailed explanation of the fine integration of EDiT with the distributed training framework to highlight our method's improvement, and provide better transparency to readers less familiar with large scale distributed training. We also acknowledge that our original paper had some misinterpretations and lacked some related experiments, so we have made the following revisions to the paper:
> 1. Emphasized the integration of our method with model sharding strategies, as well as the superiority of our approach in synchronizing sharded parameters layer by layer during forward computation.
> 2. Added more introductions to CO2 and SlowMo in Section 2 (Related Work), highlighting the distinctions and advantages of our method.
> 3. Supplemented Section 4.3 and A.3.2 with speed comparisons and profiling results of different methods, quantitatively and intuitively demonstrating the advantages of EDiT.

---

> ### Author Response · Authors · 2024-11-26
> **Response to Reviewer zyrD (Part 3/4)**
>
> **Q2: No comparison with CO2.**
>
> A2:  Thank you very much for your suggestion. We have added comparisons with CO2, including convergence, generalization, and acceleration. The experimental results can be found in Sections 4.2, 4.3, and A.3.2.
>
> **Q3：The impact of introduced hyper-parameters is not discussed.**
>
> A3: Thank you for your question. In this paper, $\alpha$ and $\delta$ are hyperparameters used in anomaly elimination, while $\phi$ is the hyperparameter to control the pseudo gradient clip. A larger $\alpha$ means the moving average is more closely aligned with the current situation, a larger $\delta$ means a lower probability to identify a worker as an outlier, and a larger $\phi$ means looser constraint on pseudo gradient clip. These three parameters only take effect when abnormal pseudo gradients appear, so they minimally impact the normal training process and convergence trend. Therefore, their settings are not strict.
>
> 1. Through extensive experiments, we found that $\alpha=0.02$ yields good experimental results. Therefore, this setting can be retained for other tasks as well.
> 2. It is generally considered that a point can be identified as an outlier if it deviates by three standard deviations from the mean in z-test. Hence, $\delta$ is typically set to 3.
> 3. Similar to gradient clipping in synchronous training, the setting of $\phi$ usually depends on the specific training task. When the training task is unstable, a smaller $\phi$ can be set to stabilize the training.  When the training task is relatively stable, a larger $\phi$ can be set to accelerate convergence. In our experiments, we set $\phi=10$. For other training tasks, we recommend testing $\phi=1$ and $\phi=10$, respectively.
>
> **Q4: Lack of clarity and imprecision in the writing.**
>
> A4:  We sincerely appreciate your thorough and detailed review of our paper and for pointing out the issues. Below are our responses to each of the questions you raised.
>
> **Q4.1: N models were data parallel sharded model replicas and not model-parallel splits.**
>
> A4.1: We have added a note in the third paragraph of the Introduction section stating that all workers operate in data parallelism. Additionally, we have included a schematic diagram in Figure 1 to illustrate the model sharding within each worker. Together with the detailed explanation of the models in the model shard groups and the model sync groups at the beginning of Section 3.1, and the Algorithm 1 in appendix, we hope to remove the confusion you raised. Besides, as mentioned in the last paragraph of Section 3.1, our method can also be combined with 3D parallelism, i.e., model parallelism, and the corresponding code will be open-sourced upon the acceptance of the paper.
>
> **Q4.2: line 21: "ensure training stability and improve performance." this is a bit cryptic.**
>
> A4.2:  We have revised the expression in line 21 to: "Besides, EDiT employs a pseudo gradient penalty strategy to suppress loss spikes, which ensures training stability and improves performance.", making it easier for readers to understand how the pseudo gradient penalty strategy takes effect. Additionally, in the second and third paragraphs of the Introduction, we have stated "low-quality datasets may introduce instability" and "the pseudo gradient penalty strategy can address this instability issue".
>
> **Q4.3: line 50: "Current Local SGD methods do not integrate well with modern distributed strategies". Why?**
>
> A4.3:  We have revised the statement to a clearer and less ambiguous one: "Existing Local SGD methods do not handle model sharding well". Additionally, we have added a paragraph in Section 2 (Related Work) to explain this point. In summary, most current methods are based on the DDP architecture, where a complete copy of the model is stored on each GPU, and an additional global communication step is used to synchronize all model parameters, resulting in significant memory and communication overhead. Although CO2 mentions that it can be combined with ZeRO3, this approach involves using all-gather to aggregate parameters before each forward and backward pass, making it revert to a synchronous mode. For a more detailed explanation, please refer to our previous response in Answer A1.
>
> **Q4.4: lines 73-75: the problem described is exactly the one the CO2 paper aims at solving, so saying "current local-SGD method" seems to be an overstatement here.**
>
> A4.4: We have changed "current" to "most" to prevent overstatement.
>
> **Q4.5: line 116: Can you clarify in which way DiLoCo extends SlowMo in this context?**
>
> A4.5: As you pointed out, the original statement was not accurate. We have revised the wording to emphasize that the key difference between DiLoCo and SLowMo is that DiLoCo demonstrates the suitability of the Nesterov optimizer as an outer optimizer.

---

> ### Author Response · Authors · 2024-11-26
> **Response to Reviewer zyrD (Part 4/4)**
>
> **Q4.6: line 170: What does "model synchronization" mean?**
>
> A4.6:  The term "model synchronization" here refers to the synchronization of parameters among workers within each model sync group, as already stated in the same paragraph: "parameters are synced in model sync groups, as outlined in lines 7 to 9 in Algorithm 1." For detailed procedures, please refer to Section 3.2, Figure 2, and Algorithm 2.
>
> **Q5: Intra node for model shards: What size of model can be loaded & how many intra-node model replicas does it lead to on standard GPU clusters?**
>
> A5:  In a typical 8 GPU node, we treat all GPUs in this node as a model shard group because these GPUs have highest bandwidth interconnectivitiy, meaning a model is sharded across 8 GPUs. Considering the sequence length is 4096, the vocabulary size is 79,800, the number of layer is 32, the batch size is 1, and using activation recomputation, the maximum model size that can be hosted on one node with EDiT is approximately 40B. Assuming using multiple connected clusters, we can further expand the scope of the model shard group by treating a single cluster as one model shard group, thereby extending the training scale across multiple clusters.
>
> **Q6:  line 168: Do you observe a different behavior of warm up phase?**
>
> A6:  As you mentioned, during our initial exploratory experiments, we found that the warm-up phase did not contribute to the final convergence results. However, the warm-up phase can enhance early training stability, particularly when the model is randomly initialized, effectively preventing training divergence. Therefore, we retained the warm-up phase, as described in Section 3.1, solely to improve early training stability.
>
> **Q7: Tab.1 & Fig.4: Why not experiment with A-EDiT on the in-house dataset?**
>
> A7:  We apologize for not completing this experiment before the manuscript submission deadline. We have now added the results of A-EDiT on the in-house dataset in Table 1 and Figure 4.
>
> **Q8: Fig 5.a: Can you explain the reason why A-EDiT and EDiT exhibit the same throughput in the random-straggler scenario?**
>
> A8:  Thank you for your question. In the random-straggler scenario, we randomly select a worker to slow down at each step. Given the uniformity, over a synchronization interval of 128 steps, all workers reach the synchronization point at roughly the same time. This results in minimal idle waiting periods for any worker, leading to similar throughput for both EDiT and A-EDiT.
>
> **Q9: In ablation Fig 7.a: Is it really necessary to keep Gradient Clip along with Weighted Average and Anomaly Detection?**
>
> A9:  Thank you very much for pointing out this problem. We have revised our ablation study to include results where gradient clip and weighted average were individually removed. As shown in Section 4.5, the removal of gradient clip leads to noticeable spikes, and the validation perplexity (PPL) is consistently higher than that of EDiT. This demonstrates that gradient clip plays an essential role in improving both stability and performance.

---

> ### Author Response · Authors · 2024-12-02
> **Gentle Reminder of the Discussion Deadline**
>
> Dear Reviewer zyrD,
>
> Thank you once again for your time and review of our manuscript. We understand that you have a busy schedule, and we kindly wish to remind you that the discussion deadline is approaching. If there are any remaining concerns or suggestions on how we can further improve our manuscript, we would greatly appreciate your feedback. Additionally, if you feel that we have successfully addressed the points raised in your initial review, we would be grateful if you could consider revisiting the score assigned to the paper.
>
> Sincerely,
>
> The Authors of Paper 1502

---

> ### Comment · Reviewer_zyrD · 2024-12-03
>
> >*"most previous Local SGD methods synchronize all parameters globally after optimizer updates"*
>
> This is not True for "SlowMo", which is the a reference implementation for local SGD methods (cf *"When using SlowMo, the models on the different nodes are no longer kept in sync after each iteration"* in the [official fairscale documentation]( https://fairscale.readthedocs.io/en/stable/tutorials/slowmo_ddp.html )).
>
> Moreover, the idea of aggregating the results of local models on each compute node to reach larger batch size is already present in SlowMo *(see their section 4 where they describe that the GPUs on each node is considered to be a single "worker" for local SGD on their experiment on ImageNet and WMT’16-En-De, and their `n_procs_per_node` parameter in the [official documentation](https://fairscale.readthedocs.io/en/stable/api/experimental/nn/slowmo_ddp.html))*.
>
> Given that they already use a "2 scale" data parallel strategy and synchronize the GPUs inside a node at each step, implementing ZeRO on each node to shard the parameters inside a node is straightforward. Thus, it does not seem that EDiT brings much in that regard.
>
> >"In EDiT, each GPU only needs to store sharded extra parameters"
>
> This is also the case for SlowMo (see the `slowmo_memory_efficient` parameter in the [official documentation](https://fairscale.readthedocs.io/en/stable/api/experimental/nn/slowmo_ddp.html)).
>
> >"From our perspective, CO2 is not readily compatible with ZeRO"
>
> We agree on this statement.
>
>
> We thank the authors for the additionnal experiments and the clarifications.
>
> However, **our main concern on the novelty/importance of the "sharding" strategy used here remains**, and is still the main focus of the paper. We think the paper would be stronger if the main focus was shifted towards the destabilizing effect of local updates/global ones in local SGD and the study of the pseudo gradient penalty strategy to counter it.
>
> To illustrate how straightforward this "sharding contribution" really is, we will give the example of the ["OpenDiloco" implementation]( https://github.com/PrimeIntellect-ai/OpenDiloco/blob/main/open_diloco/train_fsdp.py#L232 ). We recognize it is concurrent to your work in terms of timing, so it seems ok to not cite it, but we mean to use it to display how straighforward the main idea of EDiT is. In OpenDiloco, the Local SGD algorithm seems to use some variation of Pytorch's "Hybrid Sharding Data Parallel(HSDP)" (a 2D strategy to perform FSDP within a node and DDP across nodes, see the [official Pytorch documentation]( https://pytorch.org/tutorials/recipes/distributed_device_mesh.html )) catered for local SGD, which is *exactly* what EDiT main contribution is.
>
> Given that, we are not able to augment our score.

---

> > ### Author Response · Authors · 2024-12-03
> > **Response to Reviewer zyrD**
> >
> > Dear Reviewer zyrD,
> >
> > We sincerely appreciate your valuable feedback.
> >
> > At first, we believe that your primary concern regarding the novelty/importance of the "sharding" strategy may stem from some misunderstanding. We would like to emphasize that the key difference between our proposed EDiT method and others lies in our approach to parameter synchronization. **EDiT synchronizes sharded parameters layer by layer during the forward pass, utilizing the prefetch strategy to overlap computation and communication and utilizing the CPU-offload strategy to reduce additional memory overhead.** This point has been repeatedly emphasized in the revised manuscript as well as in the "Response to Reviewer zyrD (Part 1/4)" and "Response to Reviewer zyrD (Part 2/4)". To further assist you in understanding the distinctions of our work, we have included a portion of the important code implementation here:
> >
> > OpenDiLoCo:
> > ```
> > optimizer = DiLoCoOptimizer(**diloco_args)
> > ...
> > optimizer.step(scaler=scaler)
> > ```
> >
> > SlowMo/CO2:
> > ```
> > net = fairscale.data_parallel.SlowMoDistributedDataParallel(model, nprocs_per_node=8)
> > ...
> > optimizer.step()
> > net.perform_slowmo(optimizer)
> > ```
> >
> > EDiT:
> > ```
> > @no_type_check
> > def _pre_forward(...):
> >   ...
> >   if SYNC_MODE_CONTROL:
> >     _sync_sharded_params(state, handle)
> >   ...
> > ```
> >
> > As can be seen, the previous methods,  including the latest OpenDiLoCo, perform parameter synchronization either within or after the inner optimizer update, introducing an extra, non-overlapping communication burden. **To the best of our knowledge, we are the first to propose the layer-wise synchronization method.** The added experimental results presented in Sections 4.3 and Section A3.3 of our revised paper also demonstrate the superiority of our approach. **In summary, EDiT goes beyond a straightforward combination of ZeRO and LocalSGD.**
> >
> > >This is not True for "SlowMo", which is the a reference implementation for local SGD methods.
> >
> > It is possible that our wording led to some misunderstanding. What we intended to convey is that, upon reaching the synchronization interval (i.e., after $\tau$ steps of local updates), the Local SGD method requires global parameter synchronization, rather than synchronizing parameters at every internal iteration. At this point, previous methods performed parameter synchronization within or after the inner optimizer update, whereas EDiT synchronizes parameters during the next forward pass.
> >
> > >Moreover, the idea of aggregating the results of local models on each compute node to reach larger batch size is already present in SlowMo.
> > >..., implementing ZeRO on each node to shard the parameters inside a node is straightforward.
> >
> > We agree on this point. Using ZeRO to shard parameters within nodes based on the SlowMo node grouping is indeed a straightforward approach, and in fact, this is how OpenDiLoCo operates. **However, EDiT builds on this by further leveraging network topology, synchronizing parameters layer by layer during the forward pass. This allows for an overlap of computation and communication, which significantly differentiates it from previous methods.**
> >
> > >"In EDiT, each GPU only needs to store sharded extra parameters".
> > >This is also the case for SlowMo (see the `slowmo_memory_efficient` parameter in the official documentation).
> >
> > We agree with this statement, which is also mentioned in "Response to Reviewer zyrD (Part 2/4)"  where our statement was "they provide a memory-efficient option." However, we also highlighted the issues associated with this operation in this Response: "this operation can only shards extra parameters and outer momentum, and does not address the memory usage of the model, optimizer, and gradients. Meanwhile, this operation introduces extra communication overhead.", what we mean is that SlowMo can only shard the parameters and momentum related to the outer optimizer while leave the parameters, optimizer states, and gradients of the inner optimizer in their complete state. This deliberate sharding introduces additional communication overhead, which is clearly illustrated in the profiling results shown in Figure 9(c) of our revised manuscript. In contrast, in EDiT, the parameters of the inner optimizer are inherently sharded, so naturally, the parameters and momentum in the outer optimizer are also in a sharded state, eliminating the need for extra communication for aggregation.
> >
> > >We think the paper would be stronger if the main focus was shifted towards the destabilizing effect of local updates/global ones in local SGD and the study of the pseudo gradient penalty strategy to counter it.
> >
> > We are glad to see your interest in the pseudo gradient penalty strategy. This innovation is also one of the key points of our work, and we have provided a detailed description in the paper, including methodological details and ablation studies.
> >
> > We look forward to further discussions with you sincerely.
> >
> > The Authors of Paper 1502

---

> > > ### Comment · Reviewer_zyrD · 2024-12-03
> > >
> > > I thank the authors for their clarification.
> > >
> > > I now understand better what you meant by the "layer-wise parameter synchronization strategy" and your pre-fetching strategy, which allows to overlap some communication with the computations. This was not clear at all to me in my previous readings (for instance line 80 *"Model parameters are fully sharded along the model shard dimension and synchronized along the model sync dimension"* was a bit cryptic to me).
> > >
> > > The method seems indeed to differ from a straightforward combination of ZeRO and local-SGD. However, given that intra node communication are often orders of magnitude faster than outer node communications, and that pre-fetching during the forward pass mostly hides communications inside the same node, does your implementation leads to significant practical speedups compared to a straightforward "ZeRO+local SGD" method (that would perform these fast communications sequentially)?
> > >
> > > In any case, given that the method seems to differ from standard ones on this front, I raised my score.

---

> ### Author Response · Authors · 2024-12-03
> **Response to Reviewer zyrD**
>
> Dear Reviewer zyrD,
>
> Thank you very much for your response.
>
> During parameter synchronization in EDiT, the prefetch strategy allows for overlapping **two communication operations** with computation: one is the parameter synchronization (All-Reduce) among nodes (referred to as "model sync group" in the paper), and the other is the parameter aggregation (All-Gather) within nodes (referred to as "model shard group" in the paper). As you mentioned, the intra-node communication is often several orders of magnitude faster than the inter-node communication, thus the overhead from parameter aggregation within nodes is negligible. However, the overhead from parameter synchronization between nodes cannot be ignored. As shown in Section A.3.2, synchronizing a 1B model on two 8-GPU A100 nodes introduces an extra overhead of 160ms with the non-overlapping method (Post Local SGD), whereas EDiT introduces only an extra overhead of 19ms, which is due to the communication time of the first module that cannot be overlapped. From the experimental results in CO2, we can infer that this gap will widen further with an increase in the number of nodes, deterioration in inter-node communication, and larger model scales. The experimental results in Section 4.3 also demonstrate that EDiT can achieve high throughput even with small synchronization intervals, while the performance of Post Local SGD and DiLoCo is relatively poor. In fact, we first tried the straightforward "ZeRO+local SGD" method, but we found that the additional non-overlapping communication it introduced could affect the overall speed in practice. Therefore, we leveraged the advantages of hierarchical network topology and sharded parameters and proposed the layer-wise synchronization method.
>
> In summary, we believe that using the prefetch strategy to overlap forward computation with parameter synchronization communication is necessary.

---

### Official Review · Reviewer_8feR · 2024-11-07

**Soundness:** 3
**Presentation:** 3
**Contribution:** 2
**Rating:** 5
**Confidence:** 4

**Summary:**

The paper presents EDiT (Efficient Distributed Training) and its asynchronous variant A-EDiT, which aim to improve the efficiency of distributed training for large language models (LLMs). EDiT solves issues in existing distributed training such as communication bottlenecks, straggler delays, and limited scalability in heterogeneous environments. A pseudo-gradient penalty strategy is introduced to enhance training stability. Experimental results suggest EDiT and A-EDiT addresses the straggler issue and is more stable compared to baseline like DiLoCo.

**Strengths:**

* The paper introduces an innovative approach that combines Local SGD with asynchrony and gradient penalty,, which addresses communication overhead and resource elasticity.
* The paper rigorously evaluates EDiT and A-EDiT on multiple benchmarks, demonstrating improved performance in training speed, stability, and generalization compared to state-of-the-art methods.

**Weaknesses:**

* EDiT and A-EDiT rely on a range of hyperparameters, how should we think about and choose $\alpha, \phi, \beta$ on a new training task?
* The authors propose using gradient norm as a metric for anomaly elimination and gradient penalty. However, in llm training, gradients often have outliers on some of the examples. If we ignore examples with large gradient norms, will it create bias on training?
* It would also help if the authors can provide convergence analysis for the proposed EDiT method. Do EDiT and A-EDiT have convergence guarantee?

**Questions:**

Please see weakness section.

---

> ### Author Response · Authors · 2024-11-26
> **Response to Reviewer 8feR**
>
> Dear Reviewer,
>
> We appreciate your thorough evaluation of our paper. Below, we address your concerns and provide our responses. All revisions will be included in the forthcoming rebuttal version of the paper.
>
> **Q1: How should we think about and choose $\alpha$, $\phi$, and $\beta$ on a new training task?**
>
> A1:  Thank you for your question. In this paper, $\alpha$ is the decay of the exponential moving average in anomaly elimination, and $\phi$ is the threshold of pseudo gradient clip. $\beta=min(\frac{\phi}{G+\epsilon},1)$ is determined by $\phi$ . A larger $\alpha$ means the exponential moving average reflects the current trends more. A larger $\phi$ implies a looser constraint on pseudo gradient clip. These two hyperparameters only take effect when abnormal pseudo gradients occur and generally do not significantly impact the normal training process or convergence trends. Therefore, their settings are not stringent.
> 1. Through extensive experiments, we found that setting $\alpha = 0.02$ achieves good results, and this setting can be applied to new training tasks.
> 2. Similar to gradient clipping in synchronous training, the setting of $\phi$ is generally related to the specific training task. When the training task is unstable, a smaller $\phi$ can be set to stabilize the training. On the contrast, a larger $\phi$ can be set to accelerate convergence. In our experiments, we set $\phi = 10$. For new training tasks, we recommend testing $\phi=1$ and $\phi=10$, respectively.
>
> **Q2: Will ignoring examples with large gradient norms create bias in training?**
>
> A2: Thank you for the thoughtful question. Our conclusion regarding this issue is that it does not create training bias to ignore samples with large gradient norms.
> 1. We have analyzed the training process. In most cases, all workers are normal and have similar pseudo gradient norms, so anomaly elimination and weighted averaging do not take effect. We empirically note that only those anomalies that would otherwise cause local loss spikes in workers will trigger pseudo gradient penalty. To prevent misjudgment, we have set a warm-up process and an exponential moving average update strategy in anomaly elimination. As shown in Figure 7c, only about 0.8% tokens were flagged as outliers and excluded in the early stage of training. From Figure 4c, we can observe that as training progresses, the number of outliers decreases, resulting in a total outlier data ratio much lower than 0.8%, which has minimal impact on the overall data distribution for training. Additionally, the results in Figure 4 and Table 1 further confirm that ignoring samples with large gradient norms does not cause training bias. Notably, it has also been mentioned in some LLM technical reports that skipping data points can be a useful technique to suppress loss spikes, such as [PaLM](https://arxiv.org/abs/2204.02311).
> 2. In the paper [Self-Influence Guided Data Reweighting for Language Model Pre-training](https://arxiv.org/abs/2311.00913), the authors also found that using gradient norms for weighting in the mid-to-late stages of training can improve model performance and robustness. In fact, it can be seen as an approximate RMSProp optimization method to utilize pseudo gradient norms as a constraint metric of pseudo gradients, where the second moment is used to weight the gradients. This allows all workers to contribute equally to the update direction, thereby increasing the likelihood that the model updates along the correct direction.
>
> **Q3: Provide convergence analysis for the proposed EDiT method.**
>
> A3: Thank you very much for your constructive suggestion. We have added the convergence analysis for our proposed EDiT method. For details, please refer to "Section 5: Theoretical Analysis" in the main text and "Section A.4: Proof of Theorem 1" in the appendix.

---

> ### Author Response · Authors · 2024-12-02
> **Gentle Reminder of the Discussion Deadline**
>
> Dear Reviewer 8feR,
>
> Thank you once again for your time and review of our manuscript. We understand that you have a busy schedule, and we kindly wish to remind you that the discussion deadline is approaching. If there are any remaining concerns or suggestions on how we can further improve our manuscript, we would greatly appreciate your feedback. Additionally, if you feel that we have successfully addressed the points raised in your initial review, we would be grateful if you could consider revisiting the score assigned to the paper.
>
> Sincerely,
>
> The Authors of Paper 1502

---

> ### Author Response · Authors · 2024-12-03
> **Final Gentle Reminder - Discussion Period Ending Soon**
>
> Dear Reviewer 8feR,
>
> We wanted to reach out one final time regarding our manuscript, as the discussion period is drawing to a close very soon. We greatly value your expertise and initial feedback, and we hope our rebuttal has adequately addressed your concerns. If you have any remaining questions or if there are points that require further clarification, we would be most grateful to hear from you before the discussion period ends. Alternatively, if you feel satisfied with our responses, we would appreciate your consideration in updating the manuscript's evaluation.
>
> Best regards,
>
> The Authors of Paper 1502

---

### Author Response · Authors · 2024-11-28
**Dear Reviewers (Rebuttal)**

We would like to thank all reviewers for the time and effort they have invested in reviewing our paper. We are grateful for the constructive feedback, which has helped us improve the overall quality of our manuscript. We have carefully considered all the comments and have addressed them all in the revised manuscript as well as the official comment section for each reviewer. We also encourage all reviewers to read our official responses to the other reviewers for a more comprehensive understanding of the related issues.

Below, we summarize the strengths of our paper based on the reviewers' evaluations.

## Novelty

- `8feR` : "The paper introduces an innovative approach that combines Local SGD with asynchrony and gradient penalty, which addresses communication overhead and resource elasticity."
- `zyrD` : "The novel gradient penalty method seems effective."
- `zyrD` : "Advantages of a proper distributed implementation of a local SGD algorithm are highlighted."
- `zyrD` : "Asynchronous extension."
- `eSB4` : "It is interesting and novel idea of arranging the participating workers into a two-dimensional device mesh, i.e., model replica group and the model shard group."

## Empirical Performance

- `8feR` : "The paper rigorously evaluates EDiT and A-EDiT on multiple benchmarks, demonstrating improved performance in training speed, stability, and generalization compared to state-of-the-art methods."
- `zyrD` : "In the experiment on the in-house dataset (Fig 4.d), the unstable behavior of standard local-SGD methods is highlighted, and the gradient penalty introduced seems to be an effective solution to alleviate it."
- `zyrD` : "Section 4.3 displays the practical advantages (in terms of throughput) given by a proper implementation of local SGD algorithms using modern distributed methods (such as ZeRO-3/FSDP) when communication links between distributed workers have limited bandwidth."
- `zyrD` : "In the presence of stragglers, the advantage (in terms of throughput) of using an asynchronous extension of EDiT are highlighted in Sec.4.3."
- `eSB4` : "The experimental results also demonstrate the effectiveness of the proposed pseudo gradient penalty mechanism and asynchronous version of the proposed scheme."

## Presentation
- `eSB4` : "The presentation of this paper is fair and easy to follow."

---

### Meta-Review · Area_Chair_KkFv · 2024-12-20

**Metareview:**

The paper proposes an efficient distributed training method that combines Local SGD with (layer-wise) model sharding techniques. It also introduces a gradient penalty strategy for asynchronous updates, enabling efficient training of large language models across heterogeneous compute environments.
Reviewers found value in the method's technical contributions and experimental validation.

We hope the authors will incorporate the discussion points in the final version.

**Additional Comments On Reviewer Discussion:**

The author feedback phase was productive, converging to good agreement between all parties

---

### Decision · Program_Chairs · 2025-01-22

Accept (Poster)